# Direct radiative effects of dust aerosols emitted from the Tibetan Plateau on the East Asian summer monsoon – a regional climate model simulation

Hui Sun[1], Xiaodong Liu[1,2], and Zaitao Pan[3,4]

[1]SKLLQG, Institute of Earth Environment, Chinese Academy of Sciences, Xi'an, 710061, China
[2]CAS Center for Excellence in Tibetan Plateau Earth Sciences, Beijing, 100101, China
[3]Department of Earth and Atmospheric Sciences, Saint Louis University, St.Louis, Missouri, MO 63108, USA
[4]Key Laboratory of Meteorological Disaster, Ministry of Education, Nanjing University of Information Science and Technology, Nanjing, Jiangsu, China

*Correspondence to*: H. Sun (sunhui@ieecas.cn)

**Abstract.** While dust aerosols emitted from major Asian sources such as Taklimakan and Gobi Deserts have been shown to have strong effect on Asian monsoon and climate, the role of dust emitted from Tibetan Plateau (TP) itself, where aerosols can directly interact with the TP "heat pump" because of their physical proximity both in location and elevation, has not been examined. This study uses the dust-coupled RegCM4.1 regional climate model to simulate the spatiotemporal distribution of dust aerosols originating within the TP and their radiative effects on the East Asian summer monsoon (EASM) during both heavy and light dust years. Two 20-year simulations with and without the dust emission from TP showed that direct radiative cooling in the mid-troposphere induced by the TP locally produced dust aerosols resulted in an overall anticyclonic circulation anomaly in the low-troposphere centered over the TP region. The northeasterly anomaly in the EASM region reduces its strength considerably. The simulations found a significant negative correlation between the TP column dust load produced by local emissions and the corresponding anomaly in the EASM index ($r=-0.46$). The locally generated TP dust can cause surface cooling far downstream in eastern Mongolia and northeastern China through stationary Rossby wave propagation. Although dust from within TP (mainly Qaidam Basin) is a relatively small portion of total Asian aerosols, its impacts on Asian monsoon and climate seems disproportionately large, likely owning to its higher elevation within TP itself.

## 1 Introduction

Dust is one of the most important components of atmospheric aerosols. The main source of atmospheric dust is wind erosion in arid and semi-arid regions; it is estimated that the global atmospheric dust emission may be as high as 200–5000 Mt yr$^{-1}$ (Goudie, 1983). Because of the large amount in the atmosphere, dust effects on the environment and the climate system have attracted much attention. The inhalation of dust aerosols can harm both human and animal health; it can also

affect visibility and thus potentially increase the number of traffic accidents (Park and Kim, 2005). Dust aerosols also are

important drivers of the global climate because of their direct radiative effects on the Earth–atmosphere radiation balance and temperature (Tegen and Lacis, 1996; Miller et al., 2004). They can alter the atmospheric hydrological cycle by acting as cloud condensation nuclei and thus can modulate both the regional and global precipitation (Rosenfeld, 2001). Satellite observations have shown that dust originating from the Taklimakan Desert can travel around the globe within two weeks and alter the interaction between the atmospheric $CO_2$ and the global climate by providing nutrients to and interacting with the marine ecosystem (Uno et al., 2009).

East Asia is an important source region for dust (Zhang et al., 1996) and is home to more than half of the world's population. The lives of people in East Asia are deeply affected by the East Asian summer monsoon (EASM) and the relationship between dust aerosols and the EASM is of great interest to the scientific community. Simulations have shown that dust aerosols not only weaken the EASM (Sun et al., 2012; Guo and Yin, 2015), but can also reduce the atmospheric heat source over the Tibetan Plateau (TP) and delay the onset of the EASM (Sun et al., 2016). Aerosols, including dust aerosols, have been shown to affect the intensity of the EASM (Li et al., 2016) and variations in the EASM can modulate the spatiotemporal distribution of dust aerosols in East Asia. A recent modeling study by Lou et al. (2016) indicated that there was a negative correlation between the spring dust loading in eastern China and the East Asian monsoon.

Dust aerosols in East Asia are mainly derived from arid and semi-arid areas, including the Taklimakan Desert and the Gobi Desert. However, some studies have indicated that the TP itself may also be an important source region for dust (Zhang et al., 1996; Fang et al., 1999) and that the region is more conducive to the atmospheric transportation of dust due to its high altitude and it can interact directly with the TP thermal pump (Wu et al., 2012). However, the source and spatiotemporal distribution of dust aerosols over the TP have not been established yet. At present, there are three viewpoints about the source of dust aerosols over the TP. First, an investigation by Fang et al. (1995, 1999) showed that there exists $2047.41 \times 10^4$ km$^2$ of desert land over the TP, suggesting that the TP may be a potential source for dust. A numerical simulation by Chen et al. (2013) showed that dust aerosols were produced by local emissions over the TP in spring and winter. Second, satellite observations have shown that the aerosols over the TP are dominated by dust in spring and summer and that the dust aerosols were probably derived from the Taklimakan Desert and the Gurbantunggut Desert to the north of the TP (Huang et al., 2007; Jia et al., 2015). Third, some studies have indicated that the dust emitted from the south of the TP, such as from the Great Indian Desert, can also be transported over the Himalaya (Lau et al., 2006, 2010).

As a massive, elevated heat source, the TP can directly heat the upper troposphere. The heating anomaly over the TP has a great impact on the EASM (Yanai et al, 2006; Duan et al., 2012). Studies have shown that dust aerosols over the TP can alter the local atmospheric radiation balance, affecting both the heat source over the TP and the Asian monsoon (Lau et al., 2006, 2010; Chen et al., 2013; Sun et al., 2016). However, most previous simulation studies have focused on dust aerosols originating from the Taklimakan and Gobi deserts (Zhao et al., 2006; Wang et al., 2008; Huang et al., 2009; Sun et al., 2012) and there have been few investigations of the impact of dust aerosols emitted by the TP on the East Asian climate. The work reported here used the RegCM4.1 model to simulate climatic effects of distribution of dust aerosols surrounding the TP by performing numerical experiments with and without the emission of dust over the TP.

## 2 Numerical model and experiment design

### 2.1 RegCM4.1 model

We used the RegCM4.1 model (Regional Climate Model version 4.1), which is developed and supported by the National Center for Atmospheric Research (NCAR) and the International Center for Theoretical Physics. The model has been widely used for more than 20 years in studies of regional climatic and environmental change, especially in the simulation of the effect of aerosols on climate (Qian et al., 2003; Solomon et al., 2008; Zhang et al., 2009; Zanis et al., 2012; Ji et al., 2011, 2015; Das et al., 2015a, 2016; Mbienda et al., 2017).

The dynamic framework of RegCM4.1 core is based on the hydrostatic core of the mesoscale model MM5 (Grell et al., 1994). The radiation scheme in RegCM4.1 is the CCM3 radiation transfer process (Kiehl et al., 1996). RegCM4.1 has two land surface process schemes: (1) the biosphere atmosphere transfer scheme (BATS1e) (Dickinson et al., 1993); and (2) the Common Land Surface process module (CLM3.5) (Oleson et al., 2008). The dust cycle can only be diagnosed when BATS1e is used. The planetary boundary layer parameterization in RegCM4.1 follows the scheme of Holtslag et al. (1990) and there are three cumulus convection parameterization schemes, including Grell (Grell et al., 1993), Kuo (Anthes, 1977) and MIT-Emmanuel (Emanuel, 1991). The dust module coupled in RegCM4.1 is based on the dust emission model (DPM) of Marticorena et al. (1995) and Alfaro and Gomes (2001). It considers dust emission, dry/wet deposition and the diagnosis of the optical and radiation characteristics of dust (including long- and short-wave radiation) (Zakey et al., 2006; Zhang et al., 2009).

The coupled dust module has been described in detail in previous articles (Zakey et al., 2006; Zhang et al., 2009); so only a brief introduction is given here. There are four steps in dust parameterization. First, each model grid cell is classified as either desert or non-desert according to its soil properties (such as texture, soil type, particle size and composition) based on the United States Department of Agriculture textural classification. Second, dust emission is assumed to be a function of friction velocity ($u^*$); dust aerosols are lifted off the ground once $u^*$ exceeds a threshold value ($u_t^*(D_p)$).

$$u_t^*(D_p) = u_{ts}^*(D_p) \cdot f_{eff} \cdot f_w,  \qquad (1)$$

there $u_{ts}^*(D_p)$ depends on soil particle size($D_p$), $f_{eff}$ and $f_w$ are the correction terms for non-erodible surface roughness elements (Marticorena and Bergametti, 1995) and soil moisture content (Fecan et al., 1999), respectively.

Third, the horizontal mass flux is treated as a function of the frictional velocity and is given by:

$$dH_F(D_p) = E \cdot \frac{\rho_a}{g} \cdot u^{*3} \cdot (1 + R(D_p)) \cdot (1 - R^2(D_p)) \cdot dS_{rel}(D_p),  \qquad (2)$$

where $E$, $\rho_a$ and $g$ are the ratio of the erodible to total surface areas, the surface air density and the gravitational acceleration, respectively. $R(D_p)$ is the ratio of the threshold frictional velocity in equation (1) to the frictional velocity $u^*$ calculated within each grid cell from model prognostic surface wind and surface roughness height. $dS_{rel}(D_p)$ is the relative

surface area of a soil aggregate of diameter $D_p$ to the total surface area of soil aggregates. The vertical flux corresponding to each emission mode is calculated by:

$$F_{dust,i}(D_p) = (\frac{\pi}{6}) \cdot \rho_p \cdot D_i^3 \cdot N_i , \tag{3}$$

where $\rho_p$ is the aggregate density (2.65 g cm$^{-3}$), $D_i$ is the median diameter and $N_i$ is a function of the kinetic energy flux.

The dust particles are divided into four size bins (or modes): fine (0.01–1.0 μm), accumulation (1.0–2.5 μm), coarse (2.5–5 μm) and giant (5.0–20.0 μm). The dust transport, deposition and removal processes are given by Qian et al. (2001) and Qian and Giorgi (1999):

$$\frac{\partial \chi^i}{\partial t} = -\overline{V} \cdot \nabla \chi^i + F_H^i + F_V^i + F_C^i + S^i - R_{Wls}^i - R_{Wc}^i - D_d^i , \tag{4}$$

where $\chi$ is the dust mixing ratio, $\overline{V}$ is vector wind, and $-\overline{V} \cdot \nabla \chi^i$ is the advection, $F_H^i$ is the horizontal turbulent diffusion, $F_V^i$ is the vertical turbulent diffusion and $F_C^i$ is the convective transport. $R_{Wls}^i$ and $R_{Wc}^i$ are the wet removal terms, represented by large-scale and convective precipitation. $D_d^i$ is the dry deposition, represented by assuming fixed depositional velocities over both land and water.

The dust SW radiation is calculated using an asymmetry factor, single scattering albedo (SSA), and mass extinction coefficient based on Mie theory. Radiative flux calculation use the δ-Eddington approximate, and the optical spectrum is within 0.2–4.5 μm and is divided into 18 wavelength bands. One is in the visible band. Seven are in the ultrviolet band between 0.2–0.35 μm, and the rest are in the infrared band. Refractive index of dust for the SW window is from the Optical Properties of Aerosols and Clouds (OPAC) database (Hess et al., 1998). The dust SSA of the four bin is considered to be 0.95 (0.01–1.0 μm), 0.89 (1.0–2.5 μm), 0.80 (2.5–5.0 μm), 0.7 (5.0–20.0 μm) respectively. The corresponding extinction efficiencies are 2.45, 0.85, 0.38, 0.17, and asymmetry parameters are 0.64, 0.76, 0.81, and 0.87 respectively. In the LW domain, dust effects on emissivity (and hence absorptivity ) use the parameterization of Kiehl et al. (1996).

$$\varepsilon_{LW}(z) = 1 - \lambda^{-D \cdot k_{lwabs} \cdot b(z)} , \tag{5}$$

where D=1.66 is the diffusivity factor, $b(z)$ is the dust burden (g m$^{-2}$) of a given layer and Klwabs (m$^2$ g$^{-1}$) is the mass absorption coefficient calculated based on the Mie theory for each size bin of the relevant LW spectral windows using the LW refractive indices consistent with Wang et al., (2006).

## 2.2 Experimental design and observational data

Two numerical experiments were designed; both integrate for 20 years (excluding first two years of spin-up) using the dust-coupled RegCM4.1. The first experiment was a control experiment (CON) that used the default land use types

from the Global Land Cover Characterization dataset (Loveland et al., 2000), meaning that dust emitting sources both within and outside the TP are present. The second experiment was a sensitivity experiment (SEN) where we turned off the dust emission in the northern and northeastern TP (the deserts inside the black outline of the TP contour in Fig. 1b). To eliminate the dust emission in the TP, we set $u_t^*(D_p)$ in Eq. (2) and and $F_{dust,i}(D_p)$ in Eq. (3) to zero over the TP. All the other conditions in the sensitivity experiment were the same as in the control experiment. In order to isolate the effect of dust aerosols, only dust aerosols are included in our simulations, without considering other aerosols (such as anthropogenic or marine aerosols).

The initial and boundary conditions were taken from the NCAR/NCEP re-analysis dataset (Kalnay et al., 1996). The sea surface temperature used the National Oceanic and Atmospheric Administration sea surface temperature dataset (Reynolds et al., 2002). The topography of the TP is very complex, necessitating high spatial resolution (<60 km) to resolve localized precipitation (Gao et al., 2006). The horizontal resolution in RegCM4.1 runs was therefore set to 40 km. The simulation domain is shown in Fig. 1a and the model domain center was at (32°N, 105°E), with 240 grid cells in the west–east direction and 160 grid cells in the north–south direction, respectively. The model was run in the standard configuration of 18 vertical σ layers with the model top at 10 hPa. The integration duration for both experiments was from January 1, 1988 to December 31, 2009. The first two years were treated as the model spin-up time and only the results from the last 20 years were analyzed.

Five main types of observations were used to evaluate the simulated results of CON: (1) the monthly mean surface air temperature and precipitation, with a high resolution of 0.5×0.5°, provided by the Climate Research Unit (CRU) of the University of East Anglia (Mitchell and Jones, 2005), which was used to evaluate the simulated surface temperature and precipitation in CON; (2) the NCEP–DOE re-analysis wind field (2.5×2.5°) at 850 hPa, which was used to compare the simulated atmospheric circulation; (3) level-3 monthly mean AOD data from 2000 to 2009 obtained from the Multiangle Imaging Spectroradiometer (MISR) onboard NASA's Earth Observation System Terra satellite (http://www-misr.jpl.nasa.gov/). Since MODIS AOD has a large portion of missing data in Northwest China, MISR was used to evaluate the simulated dust AOD in CON. The effectiveness of the MISR data was investigated by Martonchik et al. (1998, 2004) and Bibi et al. (2015). (4) level-3 monthly mean pure dust AOD data under cloud free scenes (2×5°) from 2007 to 2009 obtained from Cloud-Aerosol Lidar and Infrared Pathfinder Satellite Observations (CALIPSO) (Winker et al., 2013), which was also used to evaluated the simulated dust AOD in CON. The most recent version of the L3 product included averaging of individual types of aerosols (Liu et al., 2008; Amiridis et al., 2013; Marinou et al., 2016); and (5) the AOD observed *in situ* by the Aerosol Robotic Network (AERONET), which was used to evaluate the simulated dust seasonal and interannual variation in CON.

# 3 Results of simulations

In this section we will first evaluate the CON simulation using the observed data described in the previous section. Then the results from CON and SEN experiments will be compared to determine the roles of dust aerosols generated from the TP play in the thermodynamic fields and circulations including the EASM.

## 3.1 Validation

### 3.1.1 Basic model climatology

The simulated climatology can influence the distribution of dust aerosols and their climatic effects, so CON was used to analyze the surface temperature, precipitation and atmospheric circulation at 850 hPa. The CON-simulated and CRU-observed 20-year average summer surface temperatures in East Asia are presented respectively in Figs 2a and 2b. The CRU observed temperature is >25°C in southern China, NW China and northern India, and it is <7°C over the TP. The observed north–south gradient and location of the maximum and minimum centers were captured well. The model captured the major distribution patterns of precipitation, including the reasonable SE–NW gradient and the maximum centers in southern China, the Himalaya and Indian Peninsula, with a 2–4 mmday$^{-1}$ negative bias in the Korean Peninsula and south Japan and a 2–4 mmday$^{-1}$ positive bias in the Tianshan Mountains (Figs. 2c, 2d). These simulated deviations are likely related to the cumulus convective scheme in the model (Zhang et al., 2008; Wang and Yu, 2011). RegCM4.1 captured the major characteristics of the circulations in East Asia, where southwesterlies dominate to the south side of the TP, and the location of the Indian Low is consistent with the NCEP–DOE observations (Figs. 2e, 2f).

### 3.1.2 Dust AOD comparison between simulated and MISR observed

Satellite and *in situ* observations include all types of aerosols, such as black aerosols, SO$_2$ and organic carbon; observed data for dust AOD alone are scarce. Therefore we used the MISR AOD data, as in most previous studies (e.g., Zakey et al., 2006; Zhang et al., 2009), to evaluate the spatiotemporal distribution of the dust AOD simulated by the model. Both the simulation and observations showed that the dust AOD over the TP and its surrounding areas was higher in spring and summer (Figs. 3a and 3c) and lower in autumn and winter (Figs. 3e and 3g). There were three maximum centers (>0.6) of dust AOD in spring and summer, located in the Taklimakan Desert, the Gobi Desert and the Great Indian Desert, respectively. The dust AOD over the Qaidam Basin in the NE of the TP was also >0.5 and the dust AOD over the northern TP, adjacent to the southern Taklimakan Desert, was between 0.3 and 0.5. The simulated dust AOD in these regions was reduced in autumn and winter (Figs. 3e and 3g). The MISR-observed AOD was largely consistent with the model results for the Taklimakan Desert, the Gobi Desert and the Qaidam Basin, but shows larger values in the Great Indian Desert in summer. The large value of the MISR AOD in the Sichuan Basin to the east of the TP was due to industrial emissions, which were not incorporated into our model simulation.

### 3.1.3 Dust AOD comparison between simulated and AERONET-observed

Figure 4 compares the *in situ* observed monthly mean AOD from AERONET and that simulated by RegCM4.1 at Dalanzadgad (43.6°N, 104.4°E). This is the only available AERONET site in the vicinity of the dust sources with continuous records for >10 years. The model captured the seasonal and interannual variations of AOD well, including the year with extremely high levels of dust. Observations over the TP are scarce and we could only find a site with continuous aerosol records from AERONET at Nam Co (30.77°N, 90.96°E). The seasonal variation of AOD at this site is well captured. Both the simulation and the observations showed that the dust AOD increases in spring at Nam Co (Fig. 5).

### 3.1.4 Simulated and CALIPSO-observed dust AOD comparison

While MISR and AERONET data contain all types of aerosols including those anthropogenic ones, the CALIPSO observation solely devotes to dust aerosols. Figure 6 shows that the simulated seasonal variation, center positions and magnitude of dust AOD are very consistent with those observed by CALIPSO during day and night. Both simulations and observations not only showed that dust AOD increased in spring and summer and decreased in autumn and winter, but also captured three maximum centers of dust AOD in Taklimakan, the Great Indian Desert and Qaidam Basin located in the northern TP in spring. The simulated center values were still high in summer. Besides, it is interesting to note that the simulated and observed dust AOD in the Qaidam Basin is higher at night than that during daytime (Figure 6a–6d, ), which implies that dust activities in the TP may be more prominent at night. This unusual feature may influence the radiative forcing of dust aerosol over the TP.

### 3.2 Relationship between the EASM and dust loading over the TP

To study the relationship between dust aerosols and the EASM, we used the average summer meridional wind at 850 hPa over eastern China (20–45°N, 105–122.5°E) as an EASM index, following Xie et al. (2016). This index measures the intensity of the southerly wind to the east of the TP in the lower troposphere over East Asia. It has been widely used to examine both modern and paleo-changes in the East Asian monsoon (Wang et al., 2008; Jiang and Lang, 2010). We found that the simulated difference in the EASM index (CON−SEN) and the difference in the model-simulated column dust load averaged over the TP are highly anticorrelated with a correlation coefficient $r=−0.46$ (Fig. 7). The dust aerosol increases and decreases over the TP as the index weakens and enhances respectively. Lou et al. (2016) also demonstrated a clear negative correlation between the EASM and the dust concentration over eastern China in spring. Based on the variation in the column dust load shown in Fig. 7, we chose 1994 and 2009 as heavy dust years and 2003 and 2007 as light dust years and then contrasted the dust distribution over the TP and its effects on the summer climate in the heavy/light dust years.

### 3.3 Dust aerosol distribution in heavy/light dust years

In the heavy dust years, the difference in the column dust load over the TP was greater than that in the light dust years, as expected. Two centers of maximum column dust load existed over the TP in the heavy dust years (Fig. 8a). One was located in the Qaidam Basin and the other was in the NW of the TP. The maximum values at both centers were >70 mg m$^{-2}$. However, the difference in the column dust load over the NW of the TP in the light dust years was much lower than in the heavy dust years and the central value was <25 mg m$^{-2}$. From the vertical profiles of the dust load (Figs 8c and 8d), we can see that the dust mixing ratio was higher in heavy dust years in the western TP with a central value >5 μg kg$^{-1}$. The mixing ratio was lower in the western TP in the light dust years.

### 3.4 East Asian climate anomalies in heavy/light dust years induced by dust aerosols

### 3.4.1 Temperature anomaly

Both the atmospheric heating rate and the atmospheric temperature over the TP decreased in heavy/light dust years (Fig. 9). During the heavy dust years, the dust aerosol resulted in a cooling anomaly centers in the lower troposphere (600–400 hPa) over TP core, with a cooling of >0.5 °C day$^{-1}$ due to the large dust load. These cooling anomalies resulted in a low temperature center at 500 hPa over the TP, with its central value reduced by >0.8°C. The dust aerosol load over the TP in the light dust years was much less than in the heavy dust years (Fig. 8a and 8c), so the cooling effect in the light dust years was weaker than in the heavy dust years.

The surface temperature over the TP decreased in both the heavy and light dust years (Fig. 10). In the heavy dust years, the surface temperature decreased by 0.6°C over the TP and consequently the sea–land thermal contrast was reduced. It is worth noting that the effects of TP aerosols on surface temperature were not limited to local or surrounding regions. In fact, the largest impact was in NE China, more than two thousands km away (Fig. 10a). The remote cooling is likely contributable to a cold air advection stemmed from the upstream TP aerosols to be discussed in the next subsection (Fig. 11). Similar phenomena were reported earlier from Europe for anthropogenic aerosols (Zanis 2009; Zanis et al., 2012) and from South Asia for natural aerosols (Das et al., 2015a).

### 3.4.2 Circulation

The overall effects of TP aerosols cool the troposphere surrounding the TP (Fig. 10a) and thus the land–sea thermal contrast was reduced by the dust aerosols over the TP. The atmospheric circulation anomaly induced by the dust aerosols emitted over the TP in heavy dust years shows an overall gigantic anticyclonic circulation centered over the TP with a positive anomaly (>10m) in geopotential height (Fig. 11a). The northeasterlies that run against the southwesterly monsoon is especially strong over the EASM region, which indicates that the EASM was weakened greatly. The anomaly still existed in the light dust years, but its intensity was much weaker than in the heavy dust years (Fig. 11b).

### 3.4.3 Precipitation

Figure 12 shows the simulated change in summer precipitation in East Asia induced by dust emitted over the TP in heavy and light dust years. The precipitation decreased in both the southern and the northern monsoon regions in summer in the heavy dust years as a result of weakening of the EASM (Fig. 11), and the reduction in the southern monsoon region is greater than that in the northern monsoon region. The dust aerosols also reduced precipitation in the two monsoon regions in the light dust years. This simulated suppressive effects of the dust aerosols were consistent with previously reported modeling results (Sun et al., 2012; Guo et al., 2015). Besides, precipitation in the heavy dust years reduced more than that in the light dust years in the TP, which may be due to the enhancement of descending motion induced by the strong cooling effects of dust aerosol over the TP.

### 4 Discussion

Previous research has shown that dust emitted from Asian deserts can weaken the EASM (Sun et al., 2012; Guo et al., 2015; Li et al., 2016), although the details of weakening mechanisms are still unclear. It has been suggested by some authors that the weakening of the EASM is a result of the reduction in the thermal contrast between the land and the sea induced by dust aerosols (Guo and Yin, 2015; Li et al., 2016). However, the modeling result of Sun et al. (2012) showed that the EASM is reduced by the large-scale atmospheric circulation disturbances (cyclone–anticyclone–cyclone Rossby wave train) generated by the radiative cooling of dust aerosols. In the work reported here, we considered the effects of dust aerosols emitted only within the TP itself on regional climate and found that they can also reduce the EASM significantly by weakening the heat source ("pump") over the TP and thus reduce the land–sea thermal contrast. The locally generated TP dust can cause surface cooling far downstream in eastern Mongolia and northeastern China through stationary Rossby wave propagation. our sensitivity simulations showed there was a negative correlation between the EASM and dust aerosols emitted from the TP locally.

The spring dust aerosols from the TP have a close relation with EASM. Although the cause-effect relationship is not immediately clear, the following processes are proposed as a possible mechanism of this relation based on the results in our simulation (Fig. 13). Firstly, increasing (decreasing) in dust aerosol over the TP in the heavy (light) dust years in spring can weaken (enhance) the TP heat source and thus reduce (increase) precipitation over the TP. Reduction (increase) in precipitation over the TP can also further enhance (diminish) dust emission over the TP (labelled 1 in Fig. 13). Secondly, the weakened (enhanced) TP heat source can persist from spring to summer and shrink (expand) the land-sea thermal contrast and thus weaken (enhance) the EASM. Therefore, the change of dust over the TP has an anti-correlation with the variation of EASM circulation intensity (labelled 2). Thirdly, weakened (enhanced) monsoon circulation can reduce (increase) precipitation in East Asia (labelled 3). As a result, the precipitation variation of the TP presents a positive-correlation with that of EASM.

It is worth noting that Sun and Liu (2016) demonstrated that dust emitted from Taklimakan and Gobi Deserts weakens the Asian monsoon through large-scale atmospheric circulations by 2 m s$^{-1}$ of wind at 700 hPa. This magnitude of reduction in the wind seems small compared to the values in the present study even though the emission source extent in the previous study is larger. We think both high-altitude source like the TP and low-altitude sources such as Taklimakan and Gobi Desert can weaken the EASM, but the mechanism could differ. The dust emitted from low altitude source (mainly Taklimakan and Gobi Desert) reduces the EASM mainly by the large-scale atmospheric, while the dust emitted from high-altitude source weakens the EASM by the reduction in the TP heating and in thermal contrast in the middle troposphere between the land and sea. The column dust load induced by local emissions from the TP in heavy dust years accounted for 20%   ((CON−SEN)/CON) of the total loading over the TP, its impacts on Asian monsoon and climate seems more important than the low altitude sources such as Taklimakan and Gobi Desert in East Asia. This disproportionately large impact from TP locally emitted dust is likely due to its higher elevation within TP itself so that the dust-induced cooling can more effectively weaken the TP's acting as a heat pump for the Asian monsoon. Further studies on this is rightly warranted.

One interesting finding of this study is the negative net (SW+LW) direct radiative forcing in the lower troposphere over the TP (Fig. 9). Dust direct radiative effects on the atmosphere have been reported to be predominantly positive (warming) over land areas in most of previous researches (Saeed et al., 2014; Osborne et al., 2011; Zhang et al., 2013; Banks et al., 2014; Chen et al., 2013, 2017). However, the research of Wang et al., (2011) reported strong cooling of dust aerosol in the East Asia deserts, and their research demonstrated that dust storms with the same intensity over the East Asia deserts and nearby regions may have different or even opposite direct radiative effects on the earth-atmosphere system, including the thermodynamic and dynamic structures of the earth-atmosphere system, depending on season and time (of day) of dust storm occurrence. Here we offer following possible explanations for the lower tropospheric cooling:

1. Magnitudes and even signs of dust aerosol direct radiative forcing in solar spectrum on the atmosphere are largely determined by the aerosol single scattering albedo (SSA) and to a less degree by the albedo of the underlying surface. The SSA of dust aerosols is determined by size distribution, morphology, and complex refractive index (Moosmüller et al., 2009). The chemical composition of dust affects also SSA value. For example, the SSA of fine mineral dust particles is determined by iron concentration (Moosmüller et al., 2012). Depending on their sources, the SSA of the dust aerosols can be quite different. Figure 14 shows the spatial distribution of SSA as calculated in the RegCM4.1. The SSA values are >0.9 over the TP, considerably larger than those over Taklimakan Desert, Gobi Desert, and the Great Indian Deserts where SSA is about 0.7. Past studies have shown that the radiative roles of dust aerosol plays is largely dependent on SSA. A 5% change in its value can significantly alter the magnitudes or even sign of SW radiative forcing (Solmon et al., 2008; Das et al., 2015b). Over the TP with SSA>0.9, only less than 10% of extinct solar radiation is absorbed by the dust aerosol, compared to 20–30% elsewhere, meaning that the TP dust aerosol is only 1/3 to 1/2 efficiency of those over surrounding sources.

Why is SSA of the TP dust aerosol smaller than dust elsewhere ? For a given wavelength, SSA depends dust particle size, among other factors. The dust particle size spectrum varies among different sources. Those fine dust particles (0.01–1.0 μm) over the large part of TP contribute as much as 70 % compared to 50–60% in Taklimakan Desert and some other areas (figure not shown).

2. Unlike SW forcing that occurs only in the sunshine hours, the LW cooling persists day and night. In addition, the TP dust tends to exist during night because of the nocturnal convergence driven by diurnal cycle of thermodynamics over the TP (Liu et al., 2009). This would further minimize the even weakened SW absorption (Fig S1). The combination of reduced SW heating and maintaining of LW cooling resulted in the net negative forcing in the lowest 200−300hPa of troposphere shown in Figure 9.

3. Because of the off-noon emission of dust in the TP, the zenith angle would be larger than near noontime. The investigation of Quijano et al. (2000) showed that dust direct radiative effects can become negative under specific situations like a large zenith angle. In fact many of previous studies showed local noon instantaneous irradiance when zenith angle is smaller. This relatively potential larger zenith angle may also contribute to the net negative direct forcing. In addition, many of dust studies are during dust storm events, most notably daytime on synoptic scale (~day), while our study is on climate scale (~years). The feedback processes are more complex as time scale prolongs. For example, the heating rate in Figs. 9c and 9d that looks more extensive than the net radiative flux reflects the contribution to the cooling from other processes.

4. Although Fig. 9 shows a negative radiative forcing over the TP within the lowest 200−300 hPa of the atmosphere, the net atmospheric column forcing measured by the flux difference at surface and TOA, as normally done in the literature, is still positive (Fig. S2). Thus strictly speaking, by conventional definition, the direct net radiative forcing on the whole atmospheric column is still positive on long-term average (Fig. S2), even though the TP dust results in a cooling effect in the lower troposphere.

Finally it is noteworthy that given the large variability of dust SSA among different sources in Asia, it is possible that the magnitudes or even signs of dust direct net radiative forcing on the atmosphere could vary among case studies and climate simulations over different continents. For example, the LW forcing of dust at Zhangye, China was found to be about a factor of two larger that over Saharan measured at Sal Island Cape Verde, owing to differences in the dust absorptive properties (Hansell et al., 2012).

It is very beneficial to study the impact of aerosols on climate using a RCM instead of a coarse-resolution global climate model (GCM). However, limited area RCM naturally cannot fully account for external forcing remote from the domain of interest although the lateral boundary conditions allow large-scale features to propagate into the domain. Our domain size (9600 km × 640 km) is reasonably large enough so that the weather and climate systems can have adequate spatial extent to develop within the domain, as attested by reasonable validation of wind pattern, temperature field, and

precipitation (Section 3.1). Cautions should be exercised, however, that results from regional simulations could be somewhat domain-size dependent quantitatively although main results should not be affected. It is worth mentioning that the model's internal variability could influence the results; so we compared the standard deviation of summer surface temperature and precipitation in CON with the signal induced by the dust effects (CON minus SEN) during the heavy dust

years. The signal induced by the dust is much greater than the standard deviations (figures not shown). Therefore, the dust effect reported in our simulation is significant in the heavy dust years.

Only direct radiative effects of dust were included in our model and future studies should include both direct and indirect effects. The simulated effects of dust aerosols on climate were highly sensitive to the physical characteristics of the dust aerosols, such as the SSA (Huang et al., 2014; Colarco et al., 2014; Das et al., 2015b). Therefore our results also need

to be validated by sensitivity experiments using aerosols with different properties. A recent study by Tsikerdekis et al. (2017) demonstrated that simulated dust load and induced radiation change are sensitive to the dust particle size division in the model, so further sensitivity experiments using more dust size bins would be worthwhile. In addition, many other factors can also affect the EASM, including the El Niño Southern Oscillation (Zhao et al., 2012; Liu et al., 2015), the North Atlantic Oscillation (Wu et al., 2009) and heat sources over the TP (Yanai et al., 2006; Duan et al., 2012). A recent

numerical simulation by Wang et al. (2017) showed that aerosol emissions from outside East Asian play an important part in weakening the circulation of the EASM.

## 5 Conclusions

We conducted two numerical experiments to quantify the effects of dust aerosols emitted over the TP on the EASM in heavy/light dust years using a high-resolution regional climate model. Satellite and *in situ* observations were used to

evaluate the simulated spatial distribution of dust aerosols and their seasonal and interannual variations. We analyzed the change in dust aerosols induced by emissions over the TP and their radiative effects on the EASM and summer precipitation in heavy/light dust years.

The spatiotemporal distribution of the dust AOD and their seasonal and interannual variation were captured well by the RegCM4.1 model compared with the MISR AOD and *in situ* observations from AERONET. Both the simulated and

observed AOD were higher in spring/summer and lower in autumn/winter. The simulated dust AOD was higher in the Taklimakan Desert, the Gobi Desert and the Great Indian Desert, with peak values >0.6. The simulated dust AOD in the Qaidam Basin and the northern TP were also higher. The seasonal variation in the dust AOD at Nam Co was captured well by RegCM4.1 compared to the observed aerosol AOD.

Comparative analyses of the two simulations indicated that the dust aerosols generated over the TP had a profound

influence on the EASM. The difference in the EASM index and column dust load between CON and SEN experiments are negatively correlated ($r=-0.46$). The index also weakened (enhanced) as the imported-local combined dust aerosol increased (decreased) over the TP. The net atmospheric heating rate was negative over the TP in heavy dust years as a result

of the radiative cooling effects of the dust aerosols, leading to a 0.6°C cooling in the surface and atmospheric temperatures. The land–sea thermal contrast and EASM were therefore both weakened, causing a 27% reduction in precipitation in the southern monsoon region. The dust load over the TP in the light dust years was much less than in the heavy dust years, implying large interannual variability.

*Acknowledgements*. The authors thank the two anonymous reviewers for valuable comments and suggestions. This research was jointly supported by the National Key Research and Development Program of China (2016YFA0601904), the National Natural Science Foundation of China (41405093, 41572150, and 41475085).

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

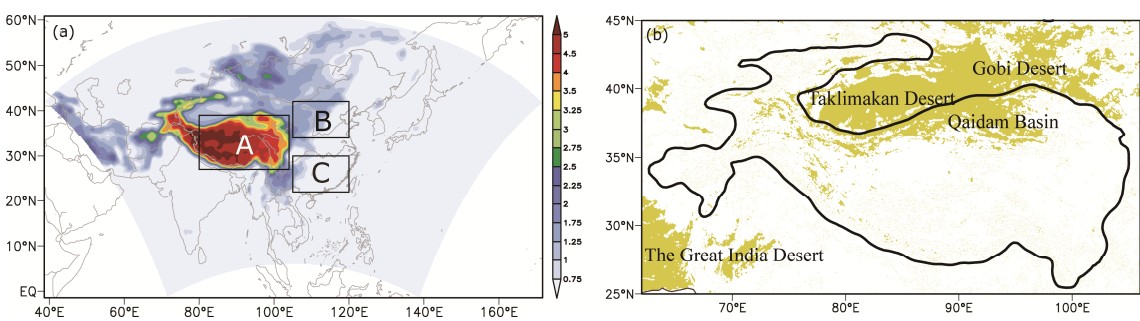

**Figure 1:** (a) Model domain and topography (units: km) and (b) dust source regions (yellow area) over the Tibetan Plateau and surrounding areas. Rectangles in (a) indicate areas Tibetan Plateau (A, 27–39°N, 80–105°E), north EASM region (B, 34–42°N, 105–120°E), and south EASM region (C, 22–30°E, 105–120°N).

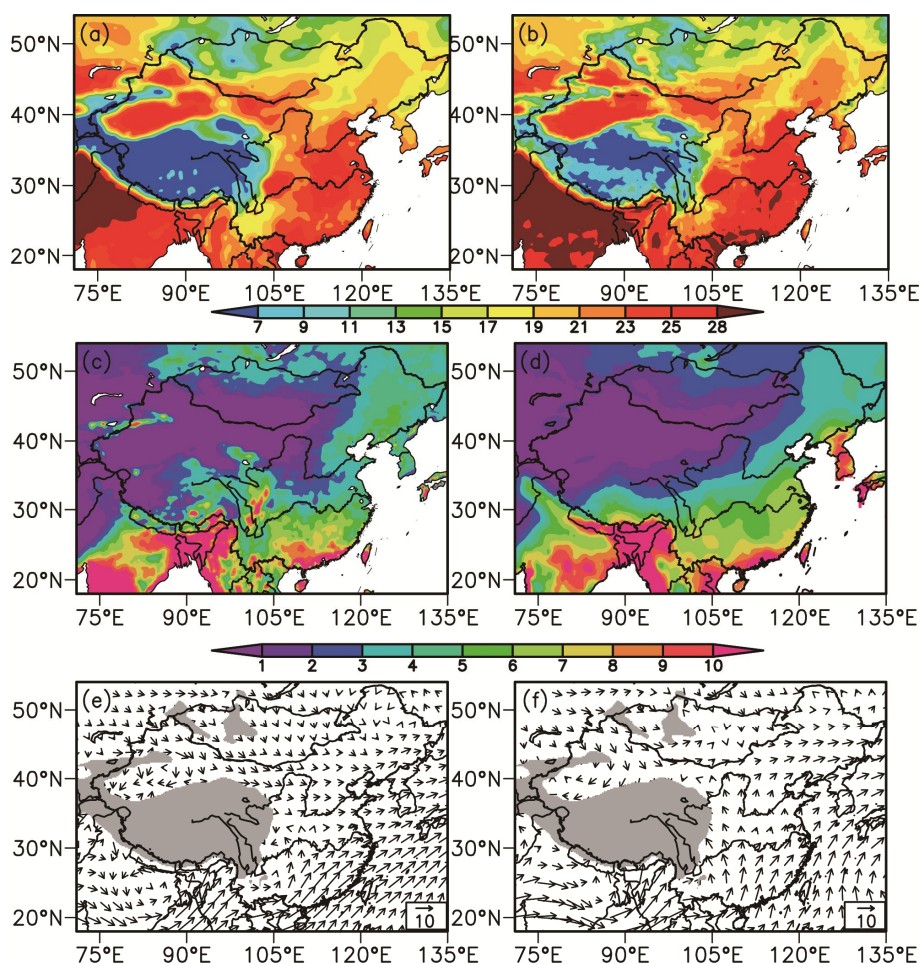

**Figure 2:** Spatial distribution of (a, b) summer surface air temperature (°C) and (c, d) summer precipitation (mm day$^{-1}$) simulated in the control experiment (left) and the CRU observations (right) for 1990–2009. The bottom two panels are wind vectors at 850 hPa simulated in (e) the control experiment and (f) the NCEP–DOE re-analysis during the summer monsoon season (June–August) averaged for 1990–2009.

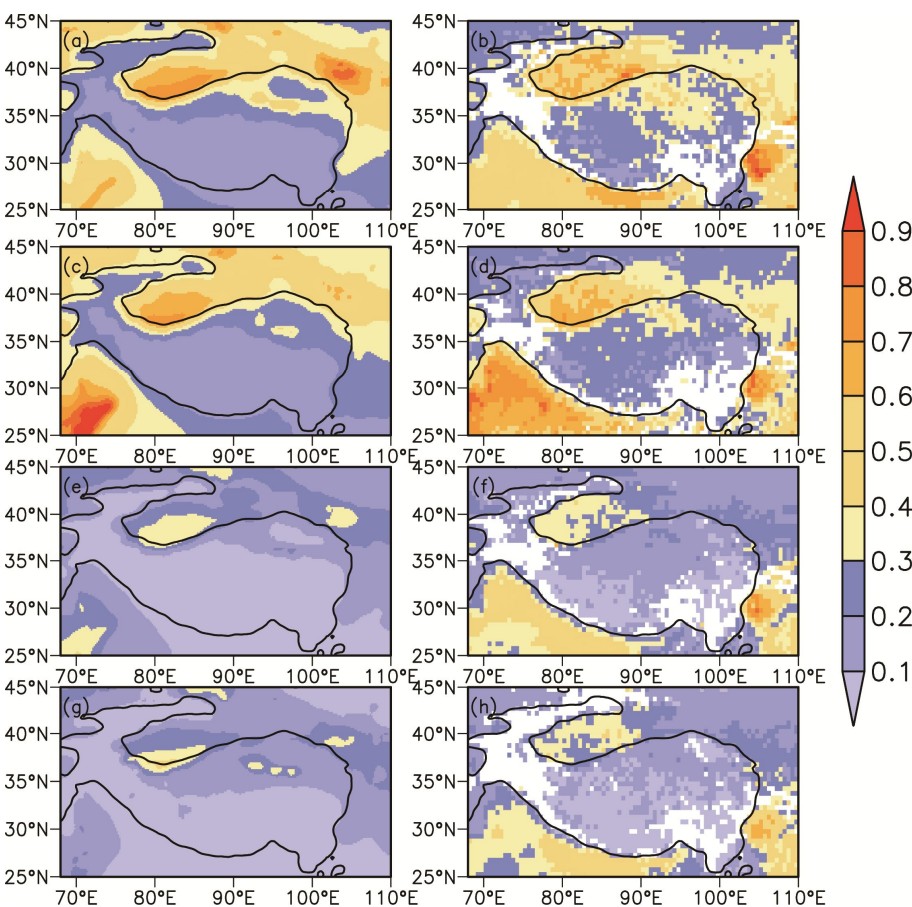

5  **Figure 3:** Spatial distribution of the dust AOD simulated by the control experiment (left panels) and the total aerosol optical depth observed by MISR at 550 nm (right panels) averaged in (a, b) spring, (c, d) summer, (e, f) autumn and (g, h) winter during the time period 2000–2009.

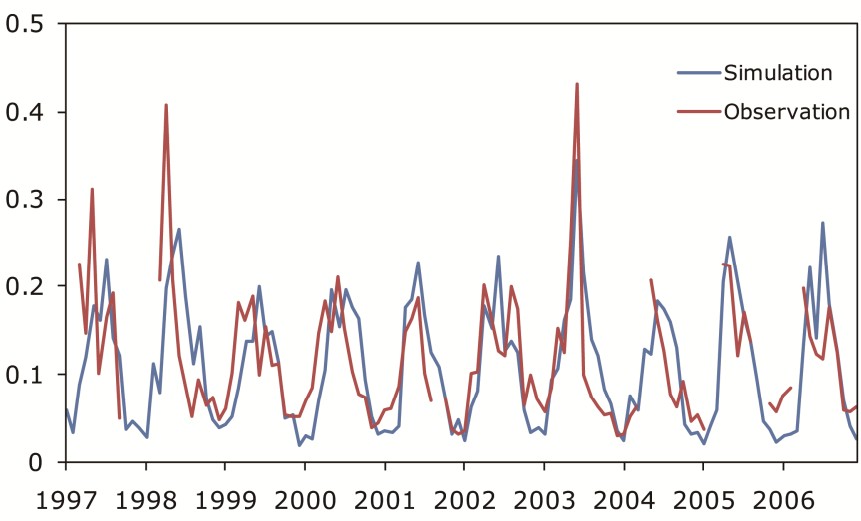

**Figure 4:** Comparison between the simulated variation of the monthly mean dust AOD in the control experiment and the

5     AERONET-observed variation of the monthly mean aerosol AOD (500 nm) at Dalanzadgad from 1997 to 2006 (*r*=0.66).

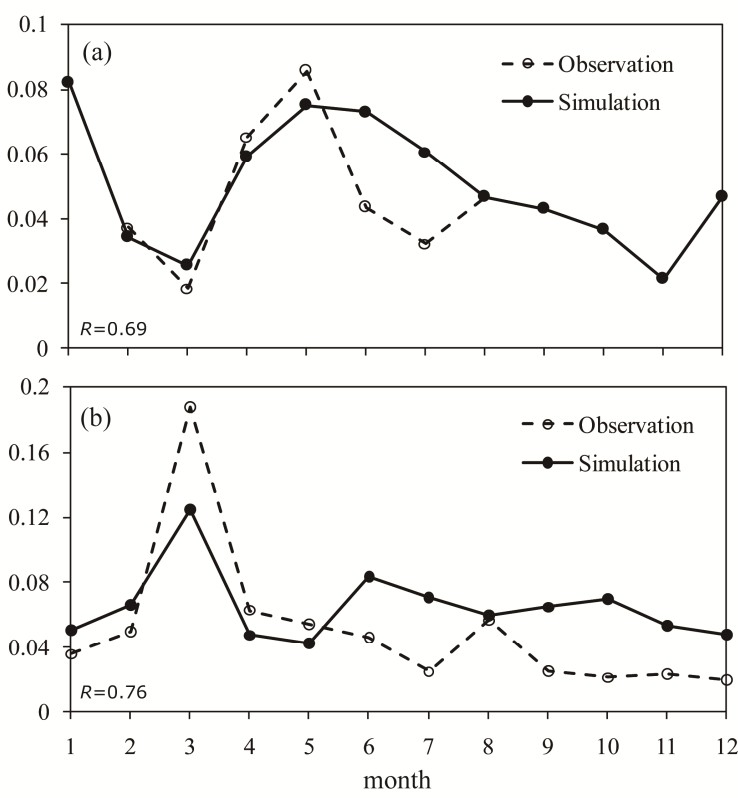

**Figure 5:** Comparison between the AERONET-observed monthly mean aerosol AOD (500 nm) at Nam Co and simulated by the control experiment at the grid near Nam Co in (a) 2007 and (b) 2009.

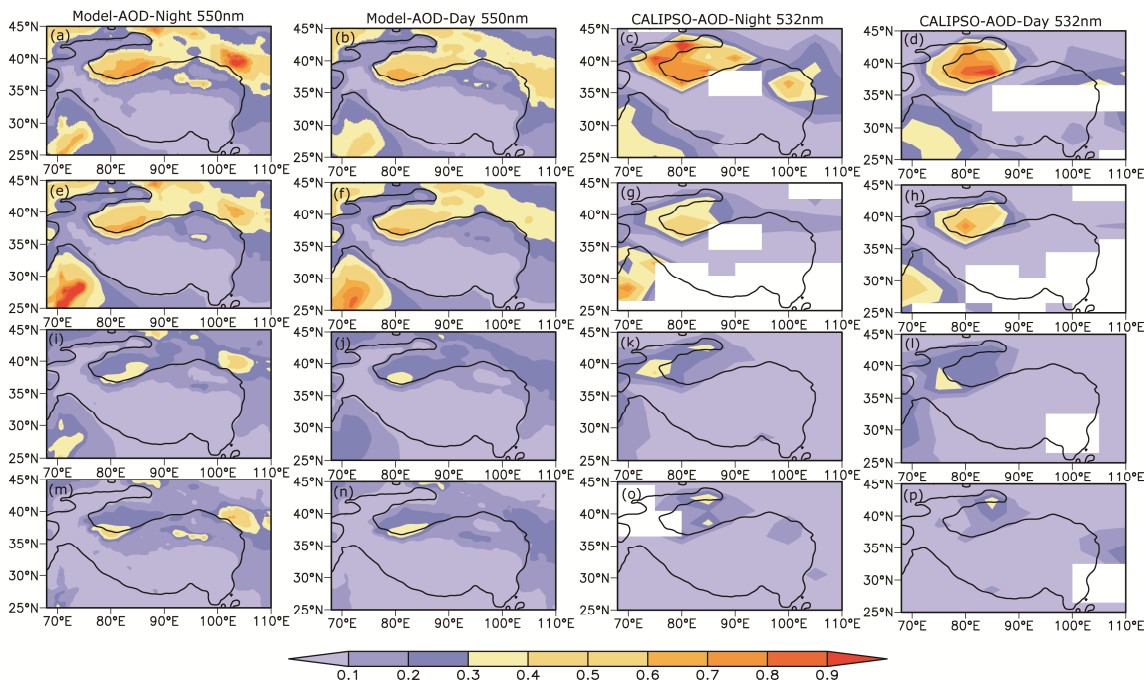

**Figure 6:** Spatial distribution of the dust AOD simulated by the control experiment at night (first column) and during daytime (second column) and the corresponding observed by CALIPSO at night (third column) and during daytime (fourth column) averaged in (a, e, i, m) spring, (b, f, j, n) summer, (c, g, k, o) autumn and (d, h, l, p) winter during the time period 2007–2009.

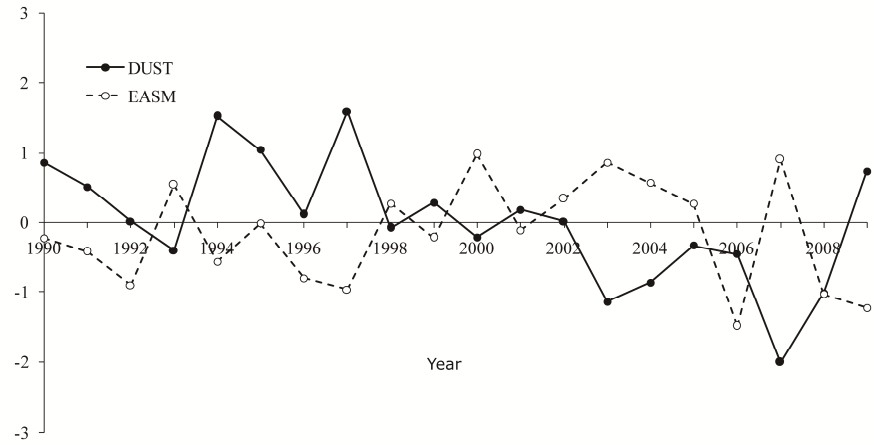

**Figure 7**: Difference (CON minus SEN) in the normalized regional mean dust column load averaged over the Tibetan Plateau (27–39°N, 80–105°E) and in the EASM index for summer during the period 1990–2009 ($r=-0.46$).

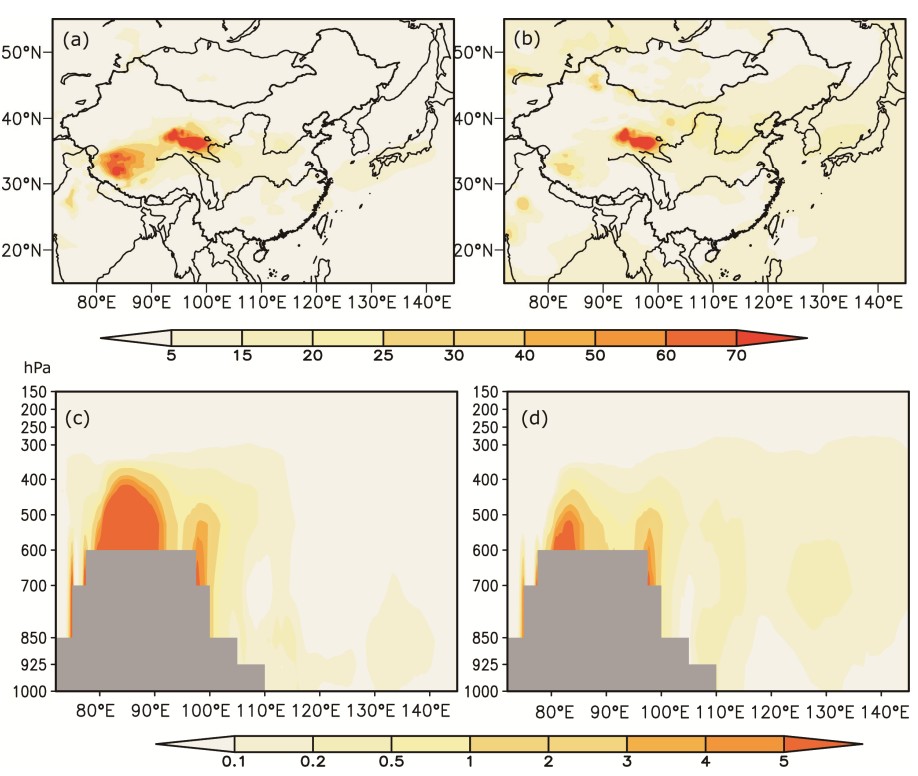

5  **Figure 8:** (a, b) Simulated differences (CON minus SEN) in the horizontal distribution of the column dust load (mg m$^{-2}$) and (c, d) the longitude–height cross-section (averaged over 32–36°N) of the dust mixing ratio (μg kg$^{-1}$) for summer in heavy (left) and light dust years (right).

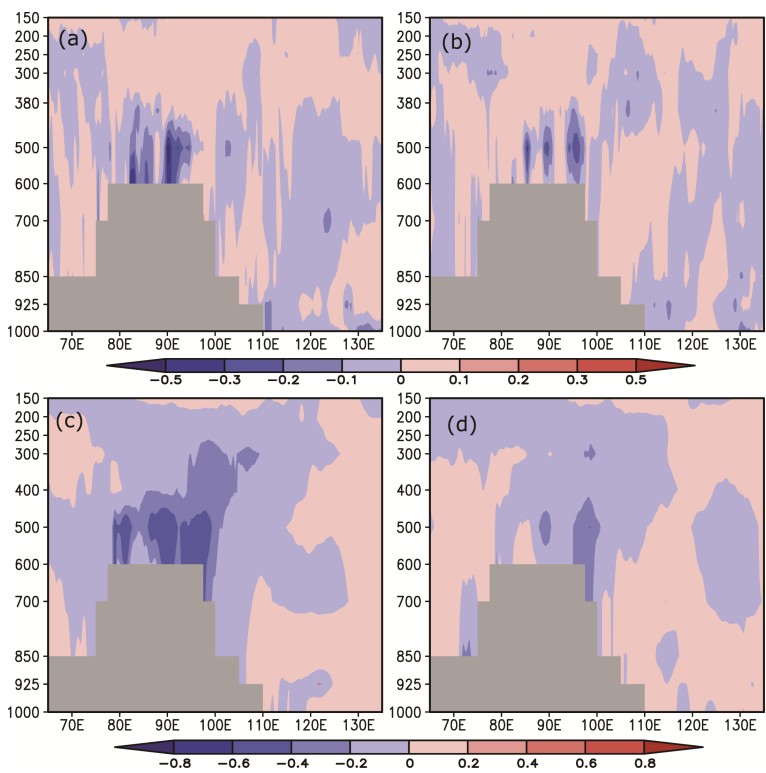

**Figure 9:** Longitudinal cross-section of the differences between CON and SEN averaged over 32–36°N in summer. (a) and (b): net radiative cooling rate (short-wave heating rate + long-wave cooling rate, °C day$^{-1}$) in heavy and light dust years respectively. (c) and (d): as (a) and (b) but for atmospheric temperature (°C).

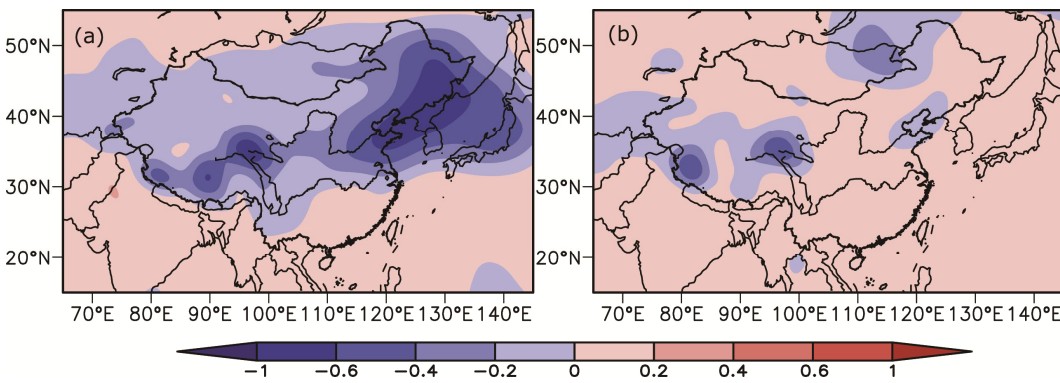

**Figure 10:** Simulated difference in summer surface air temperature between CON and SEN in (a) heavy and (b) light dust years.

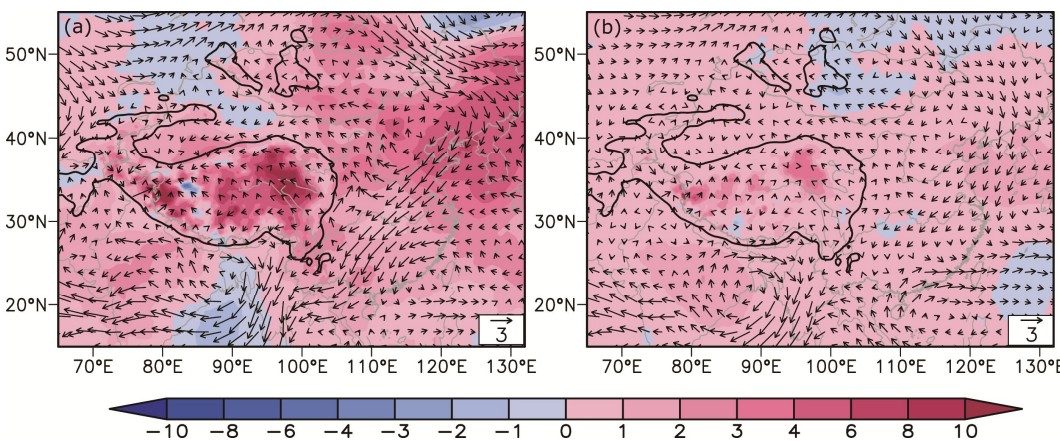

**Figure 11:** Simulated difference in atmospheric circulation at 850 hPa (vector, m s$^{-1}$) and geopotential height at 600 hPa (shaded, m) in summer   between CON and SEN in (a) heavy and (b) light dust years.

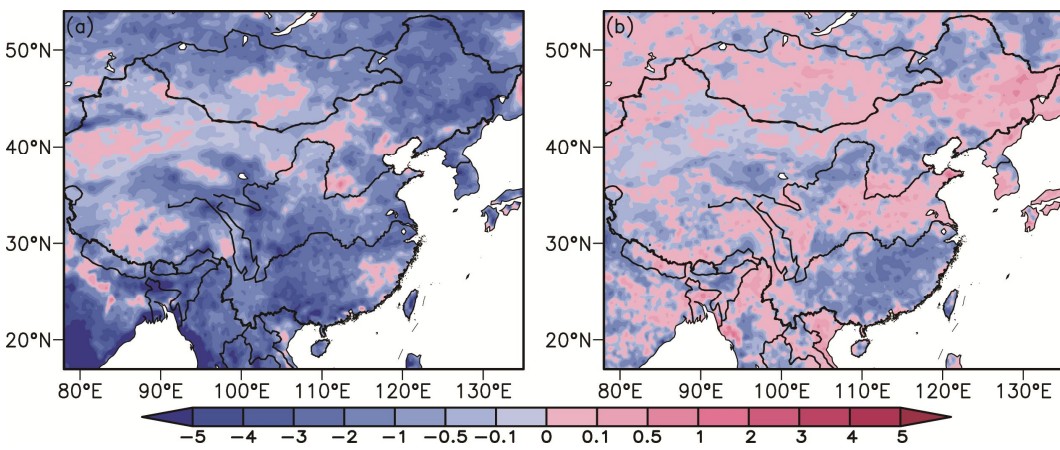

**Figure 12:** Simulated difference in summer precipitation between CON and SEN in (a) heavy and (b) light dust years.

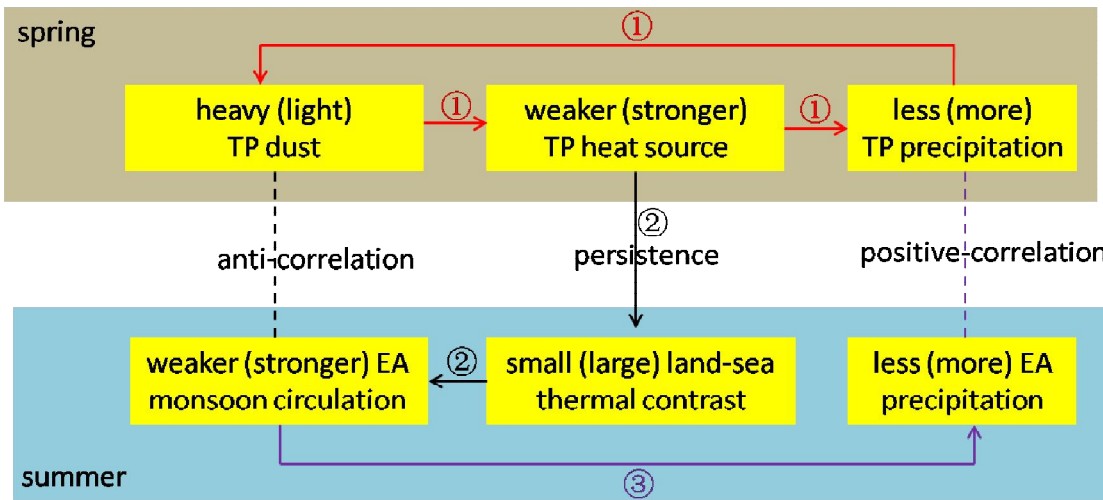

**Figure 13:** Schematic diagram showing the relation of dust aerosols emitted from the TP in spring with EASM precipitation. See text for the details.

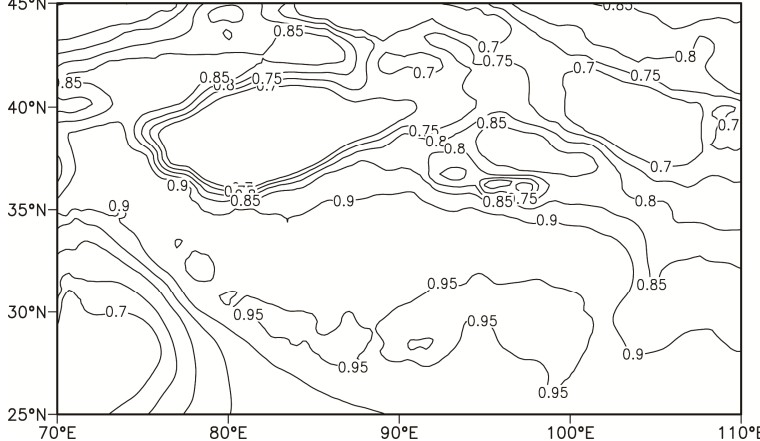

**Figure 14:** Predicted SSA in summer (CON experiment)