# Peer review of "Direct radiative effects of dust aerosols emitted from the Tibetan Plateau on the East Asian summer monsoon – a regional climate model simulation"

_Atmospheric Chemistry and Physics, 2017_

## Referee Comment (RC1) · Anonymous Referee #1 · 23 Mar 2017

This is an interesting study which investigates the direct radiative effects of dust aerosols emitted from the Tibetan Plateau on the East Asian summer monsoon with a regional aerosol-climate model. In general, it is well written and structured and there are original model results presented and discussed. However there are a number of major comments that have to be taken into consideration before acceptance of the manuscript for publication.

Comments 1) Please discuss briefly what is the added value of using a regional climate model instead of global climate model to study the impact of aerosols on climate. 2) In the discussion section the authors should also comment on the limitations of using a regional climate model to study the impact of aerosols on climate. For example, could

the authors comment if an RCM which is actually forced by lateral boundary conditions of a GCM or reanalysis can be able to provide the adequate spatial coverage for the development of atmospheric circulation feedbacks over a limited area. 3) An issue that it is not discussed at all is if the aerosol induced signal on the meteorological fields is higher than the model's internal variability. Did the authors carried out some sensitivity experiments to investigate this important issue? I think at least a few comments on this issue are necessary. This is also a part of the limitations in these simulations. 4) Page 3, lines 4-7: There are a number of other recent published studies that have looked the effect of aerosols on climate using RegCM e.g. Das et al., Clim. Dyn., 2015 and Das et al., TAC, 2016 for Asia, Zanis et al., Clim.Res. 2012 for Europe, Ji et al., Clim. Dyn., 2015 and Komkoua et al., Int. J. Clim., 2017 for Africa. 5) There is a recent study by Tsikerdekis et al., ACP (2017) testing a newly implemented 12-bin approach for RegCM which is also compared with the default RegCM4 4-bin approach used in the RegCM simulations of this work. 6) Please clarify if the RegCM simulations in this work use only dust aerosols or other aerosols as well (such as anthropogenic or marine aerosols). To my understanding the simulations include only dust particles. Hence the comparison of modelled dust AOD with AOD from satellite (MISR) and ground based (AERONET) measurements is not one to one comparison since these observations include all types of aerosols. Mind though that there are available pure dust satellite products from CALIPSO (see e.g. Amiridis et al., ACP, 2013 and Marinou et al., ACP, 2017). 7) The authors mention that in order to eliminate the dust emission in the Tibetan Plateau they replaced the land cover types of these areas with the nearby vegetated types. This change could stop the dust emission but will also change the surface albedo which itself could have an impact on radiation budget, temperature and circulation. In other words if it is like this the results do not show simply the effect of eliminating the emission of dust particles but also the effect of land cover type and albedo change. Please comment on this important issue. 8) The authors mention in Section 3.2 that "the dust aerosol increases and decreases over the TP as the EASM index weakens and enhances, respectively". Please could provide some discussion on the physical explanation for

this anti-correlation. Also provide some short description for the EASM index used. 9) The authors point is Section 3.4.1 that the effect of TP aerosols on surface temperature are not limited to the areas that the dust aerosols are locally emitted. So the effect on temperature is not solely due to local radiation imbalance from the presence of dust aerosols but also due to aerosol induced circulation changes. Similar results have been pointed for Asia by Das et al., 2015 as well as in earlier studies by Zanis, 2009 and Zanis et al., (2012) for Europe. 10) It would be helpful if the authors could also add in Figure 10 the Geopotential Height anomalies with colors to point spatially the anticyclonic circulation anomaly. 11)The discussion in Section 3.4.3 for the dust particle effect on precipitation in heavy and light dust years needs more elaboration. Maybe the authors could include a figure for precipitation similar to Figure 9 for temperature. 12) The conclusion that Figure 12 shows that the dust aerosols emitted over the TP delay the onset of the EASM is really weak since no uncertainty analysis is implemented and the differences between control and sensitivity experiments are small. I think this statement needs more elaboration and justification. 13) Since the work is focusing on summer monsoon season I think that Figure 2 (a, b , c and d) should also refer to the summer season for consistency reasons, similarly to Figures 2e and 2f .

---

## Referee Comment (RC2) · Anonymous Referee #2 · 28 Mar 2017

General comments:

The authors use an RCM in order to investigate the effect of Tibetan Plateau dust sources on the East Asian Summer Monsoon (EASM). They find that removing the desert cells in Tibet reduces precipitation and generally weakens the EASM. The subject of the study is interesting and original, the presentation is clear (but a little lacking in depth) and the flow is smooth. There are a few major points that need to be clarified or otherwise addressed before the paper is accepted.

Specific comments in order of appearance :

p.1, I.24: "dust ... accounts for about half of all the aerosols". By mass? This is not

supported from Table 1 of the Chin et al. 2002 reference.

p.5, I.10: The alleged supremacy of MISR over MODIS is justified based on only one paper which itself is based on only one AERONET station. The better agreement of MISR AOD with the specific station cannot be considered representative, see for example Bibi et al., 2015 (http://dx.doi.org/10.1016/j.atmosenv.2015.04.013) who show that MISR compares better with AERONET at two stations, and MODIS at the other two stations of the Indo-Gangetic plains. I wouldn't exclude MODIS from the analysis.

p.6, I.20: It would be nice to include some statistics (correlation coefficient, bias, etc) in Figs. 4 and 5

p.7, I.13 and Fig.8: The widespread aerosol-induced cooling is quite impressive, but also raises questions. In much of the literature the direct radiative effect of dust is predominantly positive (warming) over land areas and becomes negative in specific situations like a large zenith angle (e.g. Quijano et al., 2000, J. Geophys. Res, 105(D10), 12207-12219). Specifically over Tibet, Chen et al. (2006) (reference in manuscript) show net aerosol warming. I would suggest that the authors explore more their derived aerosol cooling and provide information on the reasons behind this behaviour. For example, is the LW cooling from dust particles so much larger than the SW warming? How much less absorbing in SW is Tibetan dust compared to dust from other locations? What are the optical properties of the dust emitted by Tibet?

p.7, l.18: It would be interesting to see why the dust generates this downward motion.

Fig.9: How does the Tibetan dust cause cooling over central India only during the light dust years? I'm afraid that using heavy and light dust years introduces aerosolunrelated interannual variability that complicates the picture. It would be much better if it were possible to tweak the dust productivity directly (please see below).

p.7, I.27 and Fig 10: As mentioned also by referee #1, the anticyclonic activity might be better visualized through geopotential heights.

p.8, I.15 and Fig.12: If I understand correctly, this difference in the EASM onset is rather marginal and probably circumstantial. For example if the value 5.5 were selected instead of 6, then the CON experiment shows earlier onset. The aerosol-induced delay of the EASM does not seem like a robust result.

Section 3.4.3: I think it would be interesting to show the change in precipitation with a Figure similar to Figs. 9 and 10. Also, there is no mention of precipitation changes in the north monsoon region.

A general remark: The authors focus on heavy and light dust years in order to evaluate the EASM sensitivity to aerosol emissions. Relying only on the heavy/light year classification, the problem retains the interannual variability from irrelevant factors such as the meteorological fields. Instead of (or maybe complementary to) the heavy/light year experiment, I would try reducing or increasing by specific percentages (e.g. 10%-100% in steps) the dust emission from the surface of Tibet, through modifications in the dust module. Then I would try to present the "climatological" 20-yr average change. I am not experienced with RegCM and do not know if these modifications are easy, so this is more suggestion than a requirement. This suggestion touches also on the valid problem (already pointed out by referee #1) of removing dust by substituting desert cells by vegetated ones. Except the aforementioned albedo changes, there could be other unwanted interferences to the aerodynamic resistances and land-air interactions. I would think that the tweaking of dust emission through modifications of e.g. Eqs. 2, 3 in the dust module would be a much better technique.

Technical corrections:

My rather trivial corrections are listed below

p.1, I.20: Please use "stationary" instead of "stationery"

p.1, I.26: Here "dust emission" is slightly better than "dust load".

p.1, I.29: Please use "drivers of" instead of "drivers on".

p.2, I.24: Please use "Gurbantunggut" instead of "Gubantunggut".

p.2, I.27: Please use "elevated" instead of "elevate"

p.4, Eq.4:  $\chi$  and v are never defined

p.6, I.27: Please correct "respevtively" to "respectively"

p.9, I.22: Please use "spatiotemporal" instead of "spatiotemporal spatial"

---

## Author Comment (AC1) · 26 May 2017

Response to **Anonymous Referee #1** concerning the paper "Direct radiative effects of dust aerosols emitted from the Tibetan Plateau on the East Asian summer monsoon – a regional climate model simulation" (**http://www.atmos-chem-phys-discuss.net/acp-2017-55/**)

We are very thankful to the reviewer who gave insightful and constructive comments. We believe that the reviewer's comments helped us highlight some critical aspects of the paper and improve the quality of our work. We have addressed all the reviewer's comments in a point-by-point manner. In the following, the underlined *italic* texts are reviewer's comments and normal (font) texts are our responses. The bold texts have been inserted to new version of our manuscript.

*Referee #1 (Comments to Author):*

*This is an interesting study which investigates the direct radiative effects of dust aerosols emitted from the Tibetan Plateau on the East Asian summer monsoon with a regional aerosol-climate model. In general, it is well written and structured and there are original model results presented and discussed. However there are a number of major comments that have to be taken into consideration before acceptance of the manuscript for publication.*

We now add more analyses and discussions which are summarized below:

   (1) compared simulated dust AOD with pure dust observation from CALIPSO in Figure 6.

   (2) added anomaly of geopotential height in Figure 11 and updated the figure of precipitation.

   (3) carried out two new sensitive experiments to isolate the effects of changed land cover alone on the results, and discussed uncertainty brought by the method used in the manuscript.

   (4) compared the aerosol-induced signal on the meteorological fields with that from the model's internal variability.

   (5) added more discussions throughout the manuscript to clarify uncertainty of our simulations, and updated the references.

*Comments:*

*1. Please discuss briefly what is the added value of using a regional climate model instead of global climate model to study the impact of aerosols on climate.*

In the revised version, we added the following statements to characterize advantages of using RCM instead of GCM to study the impact of aerosols on climate (Page 11, Line 3–10).

**It is very beneficial to study the impact of aerosols on climate using a RCM instead of a coarse-resolution global climate model (GCM). The GCMs tend to systematically underestimate dust aerosol concentration, presumably due to their lower spatial resolution (Tegen et al., 2002; Zhang et al., 2009). The RCM simulated dust concentration, on the other hand, was closer to the observed magnitude compared with the results of global models (Sun et al., 2012). For example, previous studies showed that high-resolution RCMs had better capabilities than GCMs in simulating the effect of aerosols on Asian monsoon (Zhou and Yu, 2006; Gao et al., 2008; Ji et al., 2011). A high-resolution regional model is especially needed to capture subtle characteristics in the areas of complex terrain (Ji et al., 2011).**

*2. In the discussion section the authors should also comment on the limitations of using a regional climate model to study the impact of aerosols on climate. For example, could the authors comment if an RCM which is actually forced by lateral boundary conditions of a GCM or reanalysis can be able to provide the adequate spatial coverage for the development of atmospheric circulation feedbacks over a limited area.*

Yes, RCM's high horizontal resolution comes at the expanse of being covering a limited area. Although RCM's domain is relatively small in a global sense, the atmospheric circulation away for the domain still enter the domain via lateral boundary condition (BC). Given a properly designed BC and reasonably sized domain, RCM simulations should be able to provide adequate spatial coverage for atmospheric circulation to develop and exert feedback in the domain. The reasonably simulated in wind and temperature pattern compared with the observations did suggest the external forcing passed through the lateral boundary well and developed properly within the RCM domain. At your suggestions, the following elaboration and caution are added to Discussion Section (Page 11, Line 10–16).

**However, limited area RCM naturally cannot fully account for external forcing remote from the domain of interest although the lateral boundary conditions allow large-scale features to propagate into the domain. Our domain size (9600 km × 640 km) is reasonably large enough so that the weather and climate systems can have adequate spatial extent to develop within the domain, as attested by reasonably validation of wind pattern, temperature field, and precipitation (Section 3.1). Cautions should be exercised, however, that results from regional simulations could be somewhat domain-size dependent quantitatively although main results should not be affected.**

*3. An issue that it is not discussed at all is if the aerosol induced signal on the meteorological fields is higher than the model's internal variability. Did the authors carried out some sensitivity experiments to investigate this important issue? I think at least a few comments on this issue are necessary. This is also a part of the limitations in these simulations.*

We deeply appreciate this great comment and agree that the model's internal variability could be large, so we compared the standard deviation of summer surface temperature and precipitation in CON with signals induced by the dust effects (CON minus SEN) during heavy dust years (Figure A1). It seems that the signal induced by the dust is much greater than the standard deviations. Therefore, the dust effects in our simulation is significant. We mentioned this in the revised version as below (Page 10, Line 17–22).

[Figure]

**Figure A1:** Standard deviation of summer surface temperature (a) and precipitation (c) in CON, and differences (absolute value) of summer surface temperature (b) and precipitation (d) between CON and SEN during the heavy dust years.

**It is worth mentioning that the model's internal variability could influence the results; so we compared the standard deviation of summer surface temperature and precipitation in**

**CON with the signal induced by the dust effects (CON minus SEN) during the heavy dust years. The signal induced by the dust is much greater than the standard deviations (figures not shown). Therefore, the dust effects reported in our simulation is significant in the heavy dust years, but the cooling over central India in the light dust years may be caused by the model's internal variability.**

*4. Page 3, lines 4-7: There are a number of other recent published studies that have looked the effect of aerosols on climate using RegCM e.g. Das et al., Clim. Dyn., 2015 and Das et al., TAC, 2016 for Asia, Zanis et al., Clim.Res. 2012 for Europe, Ji et al., Clim. Dyn., 2015 and Komkoua et al., Int. J. Clim., 2017 for Africa.*

Thanks. We have cited these articles in the revised version (Page 3, Line 8–9).

*5. There is a recent study by Tsikerdekis et al., ACP (2017) testing a newly implemented 12-bin approach for RegCM which is also compared with the default RegCM4 4-bin approach used in the RegCM simulations of this work.*

Thanks. We read and cited the reference in the revised version as below (Page 11, Line 21–23).

**A recent study by Tsikerdekis et al. (2017) demonstrated that simulated dust load and induced radiation change are sensitive to the dust particle size division in the model; so further sensitivity experiments using more dust size bins would be worthwhile.**

*6. Please clarify if the RegCM simulations in this work use only dust aerosols or other aerosols as well (such as anthropogenic or marine aerosols). To my understanding the simulations include only dust particles. Hence the comparison of modelled dust AOD with AOD from satellite (MISR) and ground based (AERONET) measurements is not one to one comparison since these observations include all types of aerosols. Mind though that there are available pure dust satellite products from CALIPSO (see e.g. Amiridis et al., ACP, 2013 and Marinou et al., ACP, 2017).*

Only dust aerosols included in our simulation, and we have clarified this point in the revised version as below (Page 4, Line 26–27).

**In order to isolate the effect of dust aerosols, only dust aerosols are included in our simulations, without considering other aerosols (such as anthropogenic or marine**

**aerosols).**

We agree that these comparisons are imperfect, and are grateful for reminding us availability of the pure dust observation from CALIPSO. We added a section to compare the simulated dust AOD with those of observed by CALIPSO, and the comparison is good (Figure 6). Accordingly, we updated the data description in Section 2 (Page 5, Line 18–22), and added following figure and texts in the revised version (Page 7, Line 1–11).

**3.1.4 Simulated and CALIPSO-observed dust AOD comparison**

[Figure]

**Figure 6:** Spatial distribution of the dust AOD simulated by the control experiment (left panels) and observed by CALIPSO at night (middle) and during daytime (right panels) averaged in (a, b, c) spring, (d, e, f) summer, (g, h, i) autumn and (j, k, l) winter during the time period 2007–2009.

**While MISR and AERONET data contain all types aerosols including those anthropogenic ones, the CALIPSO observation sole devotes to dust aerosols. Figure 6 shows that the simulated seasonal variation, center positions and magnitude of dust AOD are very consistent with those observed by CALIPSO during day and night. Both simulations and observations not only showed that dust AOD increased in spring and summer and**

**decreased in autumn and winter, but also captured three maximum centers of dust AOD in Taklimakan, the Great Indian Desert and Qaidam Basin located in the northern TP in spring. The simulated center values were still high in summer. Besides, it is very interesting to noted that the observed dust AOD in the Qaidam Basin is higher at night than that during daytime (Figure 6b and 6c), which implied that dust activities in the TP may be more prominent at night. This unusual feature could cause dust radiative effects in the TP is very different from those of in other locations, which is further discussed in subsequent section of 3.4.1.**

*7. The authors mention that in order to eliminate the dust emission in the Tibetan Plateau they replaced the land cover types of these areas with the nearby vegetated types. This change could stop the dust emission but will also change the surface albedo which itself could have an impact on radiation budget, temperature and circulation. In other words if it is like this the results do not show simply the effect of eliminating the emission of dust particles but also the effect of land cover type and albedo change. Please comment on this important issue.*

We deeply appreciate this great comment. We agree that our method is imperfect and uncertainty exists in the results. To address your concerns, we carried out two additional sensitive experiments to evaluate effects of the changed land cover. Dust cycle in the two experiments was turned off, but the land cover was changed to become similar to the modification in SEN. The differences between the two experiments only included effects of the changed land cover. The results showed that change (from desert to vegetated land) brought about 0.4 °C warming (Figure A2 and Figure A3), and this warming effect is weaker than the combined cooling effects (–0.6°C) induced by the dust aerosols and the changed land cover. Besides, we have noted that Li and Xue (2010) demonstrated that land cover change from vegetated land to bare ground (mainly desert) in the TP decreased the radiation absorbed by the surface and resulted in weaker surface thermal effects, which means that effect of land cover change from desert to vegetated land may also cause warming effects. This is opposite to the cooling effects induced by the dust aerosols over the TP in our simulation, and it may partly offset the dust aerosols induced cooling effects. However, our results showed that the cooling signal was not changed, and it may even be underestimated in the heavy and light dust years. Therefore, actual cooling should be stronger than the simulated. We will try a new better method in our future work. We mentioned the uncertainty brought by the method and discussed the influences on the results in the revised version as below (Page 10, Line 22–34, and Page 11, Line 1–2).

[Figure]

**Figure A2:** Longitudional-height cross-section of air temperature anomaly induced by the land cover changed averaged over 32–36°N in summer in heavy (a) and light (b) dust years.

[Figure]

**Figure A3:** Summer surface temperature change induced by the land cover change in (a) heavy and (b) light dust years.

**Besides, the results could also include the role of changed land cover in addition to the role of dust aerosols because turning off dust emission in the TP was through modifying underlying surface types. Hence, we carried out two additional sensitive experiments to isolate effects of the changed land cover alone. The dust cycle in the two experiments was turned off, but the land cover was changed to one similar to the modification in SEN. The differences between them only included effects of the change in land cover. The results showed that the change (from desert to vegetated land) brought about 0.4 °C warming (figures not shown), and this warming effects is weaker than the combined cooling (–0.6°C) induced by the dust aerosols and the changed land cover together. Besides, it is interesting to note that Li and Xue (2010) had demonstrated that the land cover change from vegetated land to bare ground (mainly desert) in the TP decreased the radiation absorbed by the surface and resulted in weaker surface thermal effects, which means that the effect of land cover change from desert to vegetated land may also cause warming effects. This is**

**opposite to the cooling effect induced by the dust aerosols over the TP in our simulation, and the warming may partly offset the dust aerosols-induced cooling effect. However, our results showed that the signal of cooling effects was not changed, although it may even be underestimated. Therefore, actual cooling should be stronger than the simulated value. Hence, the reported dust effects also need be evaluated by using a refined way in the future.**

*8. The authors mention in Section 3.2 that "the dust aerosol increases and decreases over the TP as the EASM index weakens and enhances, respectively". Please could provide some discussion on the physical explanation for this anti-correlation. Also provide some short description for the EASM index used.*

We added some short description for the EASM index in Section 3.2 in the revised version as below (Page 7, Line 14–15).

**This index measures the intensity of the southerly wind to the east of TP in lower troposphere above East Asia.**

We added following schematic diagram and texts for physical explanation of the anti-correlation in the revised version (Page 9, Line 16–25).

[Figure]

**Figure 13** Schematic diagram showing the relation of dust aerosols emitted from the TP in spring with EASM precipitation

**The spring dust aerosols from the TP have a close relation with EASM. Although the cause-effect relationship is not immediately clear, the following processes are proposed as a**

**possible mechanism of this relation based on the results in our simulation (Fig. 13). Firstly, increasing (decreasing) in dust aerosols over the TP in the heavy (light) dust years in spring can weaken (enhance) the TP heat source and thus reduce (increase) precipitation over the TP. Reduction (increase) in precipitation over the TP can also further enhance (diminish) dust emission over the TP. Secondly, the weakened (enhanced) TP heat source can persist from spring to summer and shrink (expand) the land-sea thermal contrast and thus weaken (enhance) the EASM. Therefore, the change of dust over the TP has an anti-correlation with the variation of EASM circulation intensity. Thirdly, weakened (enhanced) monsoon circulation can reduce (increase) precipitation in East Asia. As a result, the precipitation variation of the TP presents a positive-correlation with that of EASM.**

*9. The authors point is Section 3.4.1 that the effect of TP aerosols on surface temperature are not limited to the areas that the dust aerosols are locally emitted. So the effect on temperature is not solely due to local radiation imbalance from the presence of dust aerosols but also due to aerosol induced circulation changes. Similar results have been pointed for Asia by Das et al., 2015 as well as in earlier studies by Zanis, 2009 and Zanis et al., (2012) for Europe.*

Thanks. We read and cited these references in the revised version as below (Page 8, Line 16–18).

**Similar phenomena were reported earlier from Europe for anthropogenic aerosols (Zanis 2009; Zanis et al., 2012) and from South Asia for natural aerosols (Das et al., 2015a).**

*10. It would be helpful if the authors could also add in Figure 10 the Geopotential Height anomalies with colors to point spatially the anticyclonic circulation anomaly.*

Thanks for the suggestion. We updated this figure and rephrased this part. Please see Section 3.4.2 in our revised version (Page 8, Line 20–25).

[Figure]

**Figure 11:** Simulated differences in atmospheric circulation at 850 hPa (vector, m s$^{-1}$) and geopotential height at 600 hPa (shaded, m) in summer between CON and SEN in (a) heavy and (b) light dust years.

The overall effects of TP aerosols cool the troposphere surrounding the TP (Fig. 10a) and thus the land–sea thermal contrast was reduced by the dust aerosols over the TP. The atmospheric circulation anomaly induced by the dust aerosols emitted over the TP in heavy dust years shows an overall gigantic anticyclonic circulation centered over the TP **with a positive anomaly (>10m) in geopotential height** (Fig. 11a). The northeasterlies that run against the southwesterly monsoon is especially strong over the EASM region, which indicates that the EASM was weakened greatly. The anomaly still existed in the light dust years, but its intensity was much weaker than in the heavy dust years (Fig. 11b)**.**

*11. The discussion in Section 3.4.3 for the dust particle effect on precipitation in heavy and light dust years needs more elaboration. Maybe the authors could include a figure for precipitation similar to Figure 9 for temperature.*

Thanks for the suggestion. We updated this part with a horizontal distribution figure for precipitation change similar to temperature and rephrased this part. Please see Section 3.4.3 in the revised version (Page 8, Line 26–31, and Page 9, Line 1–3).

[Figure]

**Figure 12:** Simulated difference in summer precipitation between CON and SEN in (a) heavy and (b) light dust years.

Figure 12 shows the simulated change in summer precipitation in **East Asia** induced by dust emitted over the TP in heavy and light dust years. The precipitation decreased in **both the southern and the northern monsoon regions** in summer during heavy dust years as a result of weakening EASM (Fig. 11), **and the reduction in the southern monsoon region is greater than that in the northern monsoon region**. The dust aerosols also reduced precipitation in the two monsoon regions in the light dust years. The simulated suppressive effects of the dust aerosols were consistent with previously reported modeling results (Sun et al., 2012; Guo et al., 2015). **Besides, precipitation in the heavy dust years reduced more than that in the light dust years in the TP, which may be suppressed by the enhancement of descending motion induced by the strong cooling effects of dust aerosols over the TP.**

*12. The conclusion that Figure 12 shows that the dust aerosols emitted over the TP delay the onset of the EASM is really weak since no uncertainty analysis is implemented and the differences between control and sensitivity experiments are small. I think this statement needs more elaboration and justification.*

Yes. Results of this part is not very robust. Since it is off the major focus of this paper, we delete this part in our revised version. We will explore this in the future.

*13. Since the work is focusing on summer monsoon season I think that Figure 2 (a, b , c and d) should also refer to the summer season for consistency reasons, similarly to Figures 2e and 2f .*

All the figures in Figure 2 are summer average. We rephrased the caption of Figure 2 as below (Page 20, Line 6).

**Figure 2:** Spatial distribution of (a, b) **summer** surface air temperature (°C) and (c, d) **summer** precipitation (mm day$^{-1}$) simulated in the control experiment (left) and the CRU observations (right) for 1990–2009. The bottom two panels are wind vectors at 850 hPa simulated in (e) the control experiment and (f) the NCEP–DOE re-analysis during the summer monsoon season (June–August) averaged for 1990–2009.

**References**

Amiridis, V., Wandinger, U. Marinou, E. Giannakaki, E., Tsekeri, A., Basart, S., Kazadzis, S., Gkikas, A., Taylor, M., Baldasano, J., Ansmann, A.: Optimizing CALIPSO Saharan dust retrievals, Atmos. Chem. Phys., 13, 12089–12106, doi: 10.5194/acp-13-12089-2013, 2013.

Gao, X. J., et al.: Reduction of future monsoon precipitation over China: comparison between a high resolution RCM simulation and the driving GCM. Meteor Atmos Phys, 100, 73–86, 2008.

Guo, J. and Yin, Y.: Mineral dust impacts on regional precipitation and summer circulation in East Asia using a regional coupled climate system model, J. Geophys. Res., 120, 10378–10398, doi: 10.1002/2015JD023096, 2015.

Ji, Z. M., Kang, S. C., Zhang, D. F., Zhu, C. Z., Wu, J., and Xu, Y.: Simulation of the anthropogenic aerosols over South Asia and their effects on Indian summer monsoon, Clim Dyn, 36: 1633–1647, 2011

Li, Q., and Xue Y. K.: Simulated impacts of land cover change on summer climate in the Tibetan Plateau, Envrion. Res. Lett., 5, 015102, doi: 10.1088/1748-9326/5/1/015102, 2010.

Marinou, E., Amiridis. V., Binietoglou, I. Solomos, S., Proestakis. E., Konsta, D., Tsikerdekis, A., Papagiannopoulos, N., Vlastou, G., Zanis, P., Balis, D., Wandinger, U., Ansmann, A.: 3D evolution of Saharan dust transport towards Europe based on a 9-year EARLINET-optimized CALIPSO dataset, Atmos. Chem. Phys. Discuss. doi: 10.5194/acp-2016-902, 2016.

Sun, H., Pan, Z. T., and Liu, X. D.: Numerical simulation of spatial-temporal distribution of dust aerosol and its direct radiative effects on East Asian climate, J. Geophys. Res., 117(D13), 110–117, doi: 10.1029/2011JD017219, 2012.

Tegen, I., Harrison, S. P., Kohfeld, K., Prentice, I. C., Coe, M., and Heimann, M.: Impact of vegetation and preferential source areas on global dust aerosol: Results from a model study, J. Geophys. Res., 107(D21), 4576, doi:10.1029/2001JD000963, 2002.

Tsikerdekis, A., Zanis, P., Steiner, A. L., Solmon, F., Amiridis, V., Marinou, E., Katragkou, E., Karacostas, T., Foret, G.: Impact of dust size parameterizations on aerosol burden and radiative forcing in RegCM4, Atmos. Chem. Phys., 17, 769–791, doi:

10.5194/acp-17-769-2017, 2017.

Zanis, P.: A study on the direct effect of anthropogenic aerosols on near surface air temperature over Southeastern Europe during summer 2000 based on regional climate modeling, Ann. Geo., 27, 3977–3988, 2009.

Zanis, P., Ntogras, C., Zakey, A., Pytharoulis, I., and Karacostas, T.: Regional climate feedback of anthropogenic aerosols over Europe using RegCM3, Clim. Res., 52, 267–278, doi: 10.3354/cr01070, 2012.

Zhang, D. F., Zakey, A. S, Gao, X. J., Giorgi, F., and Solmon, F.: Simulation of dust aerosol and its regional feedbacks over East Asia using a regional climate model, Atmos. Chem. Phys., 9, 1095–1110, doi:10.5194/acp-9-1095-2009, 2009.

Zhou, T. J, Yu, R. C.: Twentieth century surface air temperature over China and the globe simulation by coupled climate models, J. Clim., 19(22): 5843-5858, 2006.

---

## Author Comment (AC2) · 26 May 2017

Response to **Anonymous Referee #2** concerning the paper "Direct radiative effects of dust aerosols emitted from the Tibetan Plateau on the East Asian summer monsoon – a regional climate model simulation" (**http://www.atmos-chem-phys-discuss.net/acp-2017-55/**)

We are very thankful to the reviewer who offered insightful and constructive comments. We believe that the reviewer's comments helped us highlight some critical aspects of the paper and improve the quality of our work. We have addressed all the reviewer's comments in a point-by-point manner. In the following, the underlined *italic* texts are reviewer's comments and normal (font) texts are our responses. The bold texts have been inserted to new version of our manuscript.

Referee #2 (Comments to Author)

*General comments: The authors use an RCM in order to investigate the effect of Tibetan Plateau dust sources on the East Asian Summer Monsoon (EASM). They find that removing the desert cells in Tibet reduces precipitation and generally weakens the EASM. The subject of the study is interesting and original, the presentation is clear (but a little lacking in depth) and the flow is smooth. There are a few major points that need to be clarified or otherwise addressed before the paper is accepted.*

We now add more analyses and discussions which are summarized below:

(1) compared simulated dust AOD with pure dust observation from CALIPSO in Figure 6.

(2) added anomaly of geopotential height in Figure 11 and updated the figure of precipitation.

(3) carried out two new sensitive experiments to isolate the effects of changed land cover alone on the results, and discussed the uncertainty brought by the method used in the manuscript.

(4) compared the aerosol-induced signal on the meteorological fields with that from the model's internal variability.

(5) added more discussions throughout the manuscript to clarify uncertainty of our simulations, and updated the references.

*Specific comments in order of appearance :*
*p.1, l.24: "dust ... accounts for about half of all the aerosols". By mass? This is not supported from Table 1 of the Chin et al. 2002 reference.*

Yes. Dust aerosol has the largest emissions (1500 Tg/yr) and abundance of mass (32.2 mg/m$^2$) compared to other aerosols (IPCC, 1994). We overstated a little in the first draft. The wording was likely to lead to ambiguity; so we delete these words in our revised version (Page 1, Line 26).

*p.5, l.10: The alleged supremacy of MISR over MODIS is justified based on only one paper which itself is based on only one AERONET station. The better agreement of MISR AOD with the specific station cannot be considered representative, see for example Bibi et al., 2015 (http://dx.doi.org/10.1016/j.atmosenv.2015.04.013) who show that MISR compares better with AERONET at two stations, and MODIS at the other two stations of the Indo-Gangetic plains. I wouldn't exclude MODIS from the analysis.*

MODIS AOD has a large portion of missing data in Northwest China; so we only used the data from MISR. Our statement was inaccurate and thus we rephrased these sentences as below (Page 5, Line 14–22). We also used dust AOD from CALIPSO to evaluate our simulations (Page 7, Line 1–11).

**level-3 monthly mean AOD data during 2000 to 2009 obtained from the Multiangle Imaging Spectroradiometer (MISR) onboard NASA's Earth Observation System Terra satellite (http://www-misr.jpl.nasa.gov/). Since MODIS AOD has a large portion of missing data in Northwest China, MISR was used to evaluate the simulated dust AOD in CON. The effectiveness of the MISR data was investigated by Martonchik et al. (1998, 2004) and Bibi et al. (2015).**

*p.6, l.20: It would be nice to include some statistics (correlation coefficient, bias, etc) in Figs. 4 and 5*

Thanks. We added correlation coefficient in Figure 4 (Page 22) and Figure 5 in the revised version (Page 23).

*p.7, l.13 and Fig.8: The widespread aerosol-induced cooling is quite impressive, but also raises questions. In much of the literature the direct radiative effect of dust is predominantly positive (warming) over land areas and becomes negative in specific situations like a large zenith angle (e.g. Quijano et al., 2000, J. Geophys. Res, 105(D10), 12207-12219). Specifically over Tibet, Chen et al. (2006) (reference in manuscript) show net aerosol warming. I would suggest that the authors explore more their derived aerosol cooling and provide information on the reasons*

*behind this behaviour. For example, is the LW cooling from dust particles so much larger than the SW warming? How much less absorbing in SW is Tibetan dust compared to dust from other locations? What are the optical properties of the dust emitted by Tibet?*

[Figure]

**Figure B1** Longitudinal cross-section of the differences between CON and SEN averaged over 32–36°N in summer. (a) and (b): net short-wave heating rate (K day$^{-1}$). (c) and (d): long-wave cooling rate (K day$^{-1}$) in heavy (a, c) and light dust years (b, d) respectively.

We deeply appreciate this great suggestion. We agree that direct radiative effect of dust is predominantly positive over land areas in most of previous reports (Zhang et al., 2013; Chen et al., 2013). We further examined differences in the dust AOD between day and night over the TP using the pure dust observation from CALIPSO and found that the observed dust AOD at night is higher than that during daytime over the TP (Figure 6b and 6c), which means that the dust over the TP is more active at night than daytime. Therefore, SW warming is much weaker than LW cooling (Figure B1). We noted the positive (warming) effects reported by Chen et al., (2013), but their results included the effects of other strong absorbing aerosols such as black carbon. Besides, they did not exclude contribution of other dust sources including that from the Taklimakan and the Gurbantunggut Desert to the north of the TP and that from the Great Indian Desert to the south of the TP. The warming effects in their study may be caused by black carbon

or the dust aerosols from Taklimakan during daytime, but the cooling effects in our study is mainly caused by the dust aerosols emitted from the TP at night. We added comparison between the simulated and CALIPSO observed dust AOD in the revised version (Page 7, Line 1–11), and discussed the differences (Page 10, Line 7–16) between our simulations and those of simulated by Chen et al., (2013). The following texts and Figures 6 are added in the new version.

**3.1.4 Simulated and CALIPSO-observed dust AOD comparison**

[Figure]

**Figure 6: Spatial distribution of the dust AOD simulated by the control experiment (left panels) and observed by CALIPSO at night (middle) and during daytime (right panels) averaged in (a, b, c) spring, (d, e, f) summer, (g, h, i) autumn and (j, k, l) winter during the time period 2007–2009.**

**While MISR and AERONET data contain all types aerosols including those anthropogenic ones, the CALIPSO observation sole devotes to dust aerosols. Figure 6 shows that the simulated seasonal variation, center positions and magnitude of dust AOD are very consistent with those observed by CALIPSO during day and night. Both simulations and observations not only showed that dust AOD increased in spring and summer and decreased in autumn and winter, but also captured three maximum centers of dust AOD in Taklimakan, the Great Indian Desert and Qaidam Basin located in the northern TP in**

**spring. The simulated center values were still high in summer. Besides, it is very interesting to note that the observed dust AOD in the Qaidam Basin is higher at night than that during daytime (Figure 6b and 6c), which implies that dust activities in the TP may be more prominent at night. This unusual feature could cause dust radiative effects in the TP is very different from those of in other locations, which is further discussed in the subsequent section of 3.4.1**

**Dust direct radiative effect is reported predominantly positive (warming) over land areas in most of previous researches (Zhang et al., 2013; Chen et al., 2013), but it can become negative under specific situations like a large zenith angle (Quijano et al., 2000). It is interesting to note that it can also become negative when dust activities are mainly vigorous at night over the TP. The observed dust AOD at night is much higher than those of during daytime over the TP (Figure 6b and 6c); so the SW warming effects is quite weaker than the LW cooling effects (figures not shown). The simulations of Chen et al., (2013) included strong absorbing aerosols such as black carbon and included contribution from other dust sources such as the Taklimakan and the Gurbantunggut Desert to the north of the TP and that from the Great Indian Desert to the south of the TP; thus the warming effects of dust aerosols reported in their study is broader. In contrast, the cooling effects in our study is mainly caused by the dust aerosols emitted from the TP at night.**

*p.7, l.18: It would be interesting to see why the dust generates this downward motion.*

Geopotential heights (at 600 hPa) increased in most of land in East Asia (Figure 11a), and the downward motion induced by the LW cooling of dust over the TP is enhanced (Figure B1 C). The following texts (Page 8, Line 10–11) were added in the discussion of revised version.

**The enhanced descending motion was induced by the LW cooling effects of dust over the TP (figures not shown).**

*Fig.9: How does the Tibetan dust cause cooling over central India only during the light dust years? I'm afraid that using heavy and light dust years introduces aerosol unrelated interannual variability that complicates the picture. It would be much better if it were possible to tweak the dust productivity directly (please see below).*

Thanks for pointing this out. Interaction of aerosols with EASM is very complicated, and many

factors can influence the monsoon. Contribution of dust aerosol to the monsoon variability may be only a small part. Perhaps the dust effects is significant only in the heavy dust years or monsoon anomaly years. Therefore, we focus on the heavy and light dust years. Because of less dust aerosol in the light dust years, the cooling over central India may be caused by the model's internal variability (Figure B2 a). We explained this in the revised version as below (Page 10, Line 17–22).

**It is worth mentioning that the model's internal variability could influence the results; so we compared the standard deviation of summer surface temperature and precipitation in CON with the signal induced by the dust effects (CON minus SEN) during the heavy dust years. The signal induced by the dust is much greater than the standard deviations (figure not shown). Therefore, the dust effect reported in our simulation is significant in the heavy dust years, but the cooling over central India in the light dust years may be caused by the model's internal variability.**

_p.7, l.27 and Fig 10: As mentioned also by referee #1, the anticyclonic activity might be better visualized through geopotential heights._

Thanks for the suggestion. We updated the figure and rephrased this part, Please see Section 3.4.2 (Page 8, Line 19–25)

[Figure]

**Figure 11:** Simulated difference in atmospheric circulation at 850 hPa (vector, m s$^{-1}$) and geopotential height at 600 hPa (shaded, m) in summer between CON and SEN in (a) heavy and (b) light dust years.

The overall effects of TP aerosols cool the troposphere surrounding the TP (Fig. 10a) and thus the land–sea thermal contrast was reduced by the dust aerosols over the TP. The atmospheric circulation anomaly induced by the dust aerosols emitted over the TP in heavy dust years shows

an overall gigantic anticyclonic circulation centered over the TP **with a positive anomaly (>10m) in geopotential height** (Fig. 11a). The northeasterlies that run against the southwesterly monsoon is especially strong over the EASM region, which indicates that the EASM was weakened greatly. The anomaly still existed in the light dust years, but its intensity was much weaker than in the heavy dust years (Fig. 11b).

*p.8, l.15 and Fig.12: If I understand correctly, this difference in the EASM onset is rather marginal and probably circumstantial. For example if the value 5.5 were selected instead of 6, then the CON experiment shows earlier onset. The aerosol-induced delay of the EASM does not seem like a robust result.*

Yes. Result of this part is not very robust. Since it is off the major focus of this paper we delete this part in our revised version. We will explore this in future.

*Section 3.4.3: I think it would be interesting to show the change in precipitation with a Figure similar to Figs. 9 and 10. Also, there is no mention of precipitation changes in the north monsoon region.*

Thanks for the suggestion. We updated this part with a new figure for precipitation change similar to temperature and rephrased this part. Precipitation in the northern monsoon regions was also suppressed by the dust. Please see Section 3.4.2 (Page 8, Line 26–31, and Page 9, Line 1–3).

[Figure]

**Figure 12:** Simulated difference in summer precipitation between CON and SEN in (a) heavy and (b) light dust years.

Figure 12 shows the simulated change in summer precipitation in **East Asia** induced by dust

emitted over the TP in heavy and light dust years. The precipitation decreased **in both the southern and the northern monsoon regions** in summer in the heavy dust years as a result of weakening of the EASM (Fig. 11), **and the reduction in the southern monsoon region is greater than that in the northern monsoon region**. The dust aerosols also reduced precipitation in the two monsoon regions in the light dust years. The simulated suppressive effects of the dust aerosol were consistent with previously reported modeling results (Sun et al., 2012; Guo et al., 2015). **Besides, precipitation in the heavy dust years reduced more than that in the light dust years in the TP, which may be suppressed by the enhancement of descending motion induced by the strong cooling effects of dust aerosols over the TP.**

*A general remark: The authors focus on heavy and light dust years in order to evaluate the EASM sensitivity to aerosol emissions. Relying only on the heavy/light year classification, the problem retains the interannual variability from irrelevant factors such as the meteorological fields. Instead of (or maybe complementary to) the heavy/light year experiment, I would try reducing or increasing by specific percentages (e.g. 10%- 100% in steps) the dust emission from the surface of Tibet, through modifications in the dust module. Then I would try to present the "climatological" 20-yr average change. I am not experienced with RegCM and do not know if these modifications are easy, so this is more suggestion than a requirement. This suggestion touches also on the valid problem (already pointed out by referee #1) of removing dust by substituting desert cells by vegetated ones. Except the aforementioned albedo changes, there could be other unwanted interferences to the aerodynamic resistances and land-air interactions. I would think that the tweaking of dust emission through modifications of e.g. Eqs. 2, 3 in the dust module would be a much better technique.*

We deeply appreciate this remark and thanks for the great suggestions. On one hand, the interaction of aerosols and EASM is very complicated, and there are many factors that can influence the monsoon. Contribution of dust aerosol to the monsoon variability may be only a small part. Perhaps the dust effect is significant only in the heavy dust years or strong monsoon years. Therefore, we focused on the heavy and light dust years. We agree that the meteorological fields could influence the results, especially in the light dust years with less dust aerosol. Thus, we compared the standard deviation of summer surface temperature and precipitation in CON with the signal induced by the dust effects (CON minus SEN) during the heavy dust years (Figure B2). The signal induced by the dust is much greater than the standard deviations. Therefore, the dust effects in our simulation is significant. We mentioned this in the revised version (Page 10, Line 17–22).

[Figure]

**Figure B2** Standard deviation of summer surface temperature (a) and precipitation (c) in CON, and differences (absolute value) of summer surface temperature (b) and precipitation (d) between CON and SEN during the heavy dust years.

On the other hand, we agree that our method to turn off the dust emission in the TP is imperfect, and it will bring uncertainty to the results. Therefore, we carried out two new sensitive experiments to isolate effects of the changed land cover alone. Dust cycle in the two experiments is turned off, but the land cover is changed into one similar to the modification in SEN. The differences between them only include effects of the changed land cover. The results showed that the change (from desert to vegetated land) brought about 0.4 °C warming (Figure B3 and Figure B4), and this warming effect is weaker than the combined cooling effects (–0.6°C) induced by the dust aerosols and the changed land cover together. Besides, we have noted that Li and Xue (2010) demonstrated that the land cover change from vegetated land to bare ground (mainly desert) in the TP decreased the radiation absorbed by the surface and resulted in weaker surface thermal effects, which means that the effect of land cover change from desert to vegetated land may also cause a warming effects. This is opposite to the cooling induced by the

dust aerosols over the TP in our simulation, and it may partly offset the dust aerosol induced cooling effects. However, our results show that the signal of cooling effect is not changed and it may be even underestimated in the heavy and light dust years. Therefore, actual cooling should be stronger than the simulated value. We admit that turning off the dust emission through modifications of Eqs. 2, 3 in the dust module would be a much better choice but it would be difficult to carry out given the short time frame. It is definitely worth trying in the future. We added discussion (Page 10, Line 22–34, and Page 11, Line 1–2) about uncertainty brought by the method used in this paper. The following texts were added in the discussion of the revised version.

[Figure]

**Figure B3** Longitudinal-height cross-section of atmospheric temperature anomaly induced by the land cover changed averaged over 32–36°N in summer in heavy (a) and light (b) dust years.

[Figure]

**Figure B4** Summer temperature change induced by the land cover change in (a) heavy and (b) light dust years.

[revised manuscript text omitted]

---

## Author Response (AR2)

Manuscript No:acp-2017-55

Journal: ACP

The revised manuscript entitled "**Direct radiative effects of dust aerosols emitted from the Tibetan Plateau on the East Asian summer monsoon – a regional climate model simulation**" by Hui Sun, Xiaodong Liu, and Zaitao Pan.

We appreciate the referee's persistent effort in reviewing our manuscript and insightful comments. Especially we are very thankful for his/her checking into a number of publications in literature. We have addressed the two important issues raised by the referee #2: i) method of switching of dust emission over the TP and ii) explanations for dust's negative net direct radiative forcing on the atmosphere. The new Fig.14 and Figs. S1 and S2 are also added. By addressing these key issues along with newly added detailed clarifications in Discussion Section, we believe the manuscript is now much strengthened.

In the following, the underlined italic texts are reviewer's comments and normal (font) texts are our response. The bold texts have been inserted to new version of our manuscript.

*Referee #2 (Comments to Author):*
*Although the authors make an effort to address many points raised by the referees, they only tried to provide an explanation for the strong atmospheric LW cooling and weak SW warming through indirect means, i.e. through the examination of the day-night differences in CALIOP AOD. Also, I still cannot understand why the authors do not simply switch off the dust cycle in order to investigate its effect and try to do so through unnecessarily complicated land-cover experiments. Until these important issues are resolved, I cannot suggest publication for the manuscript. Please see my justification for this below. I have also a few other criticisms, but they are not as important.*

In the original version, we used land use type to mimic the TP dust emission switch partly because of the continuation of our previous studies testing land use change effects. Also this way needs only to alter model parameter input, avoiding change model's source code. During the previous revision, we started to implement your suggestion, but just could not finish it within the limited time frame. Now we have completed the new sensitivity experiment in which the dust cycle was switched off over the TP instead of changing the land cover in previous version, as you suggested. Now the experimental design is straightforward and the logic is easier to follow. Thank for the great suggestion.

As shown in the revised manuscript, we replaced the original results with new ones, even though the main conclusions have not changed. The negative net direct radiative forcing on

the atmosphere by the dust originated over the TP in the lower troposphere is now somewhat less extensive in the new experiment compared to the previous version, which makes the explanations for the negative atmospheric forcing quantitatively easier.

*Section 3.1.4 and Discussion*

*My additional efforts to identify cases with general dust-induced atmospheric LW cooling stronger than SW warming were fruitless. In contrast, I came up with even more studies where SW atmospheric warming is significantly larger than LW cooling (e.g. doi: 10.1002/2014JD022077, 10.5194/acp-14-3751-2014, 10.1002/qj.771 (indirectly), 10.5194/acp-17-2401-2017). Even the same authors in their earlier paper Sun et al., 2012 (DOI: 10.1029/2011JD017219) show in Fig. 9 a dust radiative effect with significant atmospheric SW warming and less significant LW cooling, as expected. In Fig. 12 of the same paper, they present dust-induced atmospheric warming. How can the radiative cooling be so prominent only for the Tibetan dust?*

Thanks again for looking into literature about signs and magnitudes of radiative forcing on the atmosphere. To address your concern, the following paragraphs are added to the Discussion Section (Page 10–11).

[revised manuscript text omitted]

*The day-night differentiation in Fig. 6 and Section 3.1.4 is not convincing. The CALIPSO night and day AOD seem rather comparable, with the exception of missing data over the Qaidam basin for the spring days. Missing AOD data does not mean small AOD. The stronger radiative effect in the LW has not been adequately justified because:*
*1) the day-night difference in the model AOD has not been shown. It is this difference that might create the stronger LW cooling, while the day-night difference of CALIPSO is not convincing. Even if it were, it would be a indirect explanation, because it is not used in the radiative transfer model. The model AOD is.*

We updated Figure 6 and plotted the simulated dust AOD during day and night respectively. The simulated dust AOD during the night over the TP was indeed higher than that during the daytime. However, the more important reason should be the fine particles (less absorbing) produced over the TP (Figure B1)

[Figure]

**Figure 6: Spatial distribution of the dust AOD simulated by the control experiment at night (first**

column) and during daytime (second column) and the corresponding observed by CALIPSO at night (third column) and during daytime (fourth column) averaged in (a, e, i, m) spring, (b, f, j, n) summer, (c, g, k, o) autumn and (d, h, l, p) winter during the time period 2007–2009.

*2) The reported CALIPSO AOD is given probably at 532 nm and the model AOD wavelength is not reported. The strong LW cooling would be better examined if the AOD in the thermal infrared were reported.*

The model AOD is at 550 nm which is closed to the CALIPSO AOD (532nm). The LW cooling may not be overestimated, but the SW warming is quite weak because of the less absorbing and more scattering dust aerosol produced over the TP (Figure B1). We added the wavelength of model AOD in the updated Figure 6 (Page 25).

*3) The optical properties of dust both in the SW and the LW are not given in the paper, even though they were requested in my previous review. The Tibetan dust might be not as absorbing in the SW as at other areas and therefore explain the weak SW warming. At least the mean area single scattering albedo would help highlight the reason for the dominant LW cooling.*

Sorry for our carelessness. We added following descriptions to clarify the dust properties and the relevant radiative calculation (Page 4, Line 14–26).

**The dust SW radiation is calculated using an asymmetry factor, single scattering albedo (SSA), and mass extinction coefficient based on Mie theory. Radiative flux calculation use the δ- Eddington approximate, and the optical spectrum is within 0.2–4.5 µm and is divided into 18 wavelength bands. One is in the visible band. Seven bands are in the ultrviolet band between 0.2–0.35 µm, and the rest are in the infrared band. Refractive index of dust for the SW window is from the Optical Properties of Aerosols and Clouds (OPAC) database (Hess et al., 1998). The dust SSA of the four bin is considered to be 0.95 (0.01–1.0 µm), 0.89 (1.0–2.5 µm), 0.80 (2.5–5.0 µm), 0.7 (5.0–20.0 µm) respectively. The corresponding extinction efficiencies are 2.45, 0.85, 0.38, 0.17, and asymmetry parameters are 0.64, 0.76, 0.81, and 0.87 respectively. In the LW domain, dust effects on emissivity (and hence absorptivity ) use the parameterization of Kiehl et al., (1996).**

$$\varepsilon_{LW}(z) = 1 - \lambda^{-D \cdot k_{lwabs} \cdot b(z)}, \tag{5}$$

**where D=1.66 is the diffusivity factor, *b(z)* is the dust burden (g m$^{-2}$) of a given layer and Klwabs (m$^2$ g$^{-1}$) is the mass absorption coefficient calculated based on the Mie theory for**

**each size bin of the relevant LW spectral windows using the LW refractive indices consistent with Wang et al., (2006).**

Thanks for your suggestion. We found that the SSA over the TP is quite larger than those of in its surrounding sources (Figure B1 and Fig. 14), which led to the weak SW warming and relative strong LW cooling in the TP.

*The authors state in their reply that "The warming effects in their study may be caused by black carbon or the dust aerosols from Taklimakan during daytime, but the cooling effects in our study is mainly caused by the dust aerosols emitted from the TP at night." Could this be verified by a two-dimensional map of the effects, similar to Fig. 8a or b?*

The dominant reason for the weak SW warming is the less absorbing dust aerosol produced over the TP (Figure B1). We clarified these statements in the revised version (Page 10-11).

*p. 10, ll. 20-22: "the dust effect reported in our simulation is significant in the heavy dust years, but the cooling over central India in the light dust years may be caused by the model's internal variability". Why? It appears that the magnitudes of cooling in the two cases are 0.6-0.8 and 0.4-0.6, respectively. Both seem quite larger than the standard deviation of Fig. B2a, unless the part of India that is not shown in Fig. B2 is characterized by larger standard deviations.*

We used the updated results of new experiments which only turned off the dust cycle without the modification of land cover, and the noise in India disappeared. Please see the Figure 10 in the updated version (Page 29).

*p. 10, ll. 24-25: "The dust cycle in the two experiments was turned off". It appears that the authors can switch off the dust cycle. This (without changing the land cover) is what I proposed in my initial review, but the authors decided to keep the land cover change. The reasons remain unclear to me. If it is possible to switch the dust cycle off, then why would one stick to the methodology of changing the Tibetan plateau cells land cover? After all, the focus of the paper is exactly the effect of the dust and I presume not the effect of the land cover type. Instead of the comparisons a) normal land cover with dust vs. altered land cover without dust and b) normal land cover without dust vs. altered land cover without dust, the authors could have only performed the more straightforward c) normal land cover with dust - normal land*

*cover without dust. I suspect that unwanted factors other than dust may influence the paper results. I would also (if possible) test the sensitivity by decreasing directly the dust emission in steps.*

We tried our best to turn off the dust emission suggested by the referee during the first revision period. We succeeded in turning off dust emission in the whole domain, but we failed to turn it off in a specific region. Therefore, we have no alternative but to indirectly contrast the role of land cover used originally by the deadline of revision. We kept trying to turn it off over the TP during the second round of revision. Fortunately, we finally succeeded. We have completed the new sensitivity experiment in which the dust cycle was switched off without changing the land cover. To eliminate the dust emission in the TP, we multiply the Eq. (2) and Eq. (3) by zero over the TP. As shown in the revised manuscript, we replaced the original results with new experimental results; the main conclusions have not changed. Please see the figures (Figure 7–12) in the updated version. We also modified the elaboration of experimental design in Page 5, Line 7.

*Related to both this point and the unexplained to me dust-induced atmospheric cooling, it is strange that in Fig. B1 of the authors reply, the enhanced LW cooling is reported as a 32-36 deg N average. I think that relatively little dust exists in these latitudes (Fig. 6 left column). Could it be that some other factor, unrelated to dust but related to the altered land cover, influences this LW cooling?*

We excluded the role of land cover in our new experiments. The enhanced dust-induced atmospheric cooling in the lower troposphere is caused by the less absorbing dust aerosol produced over the TP (Figure B1). The results in Figure 6 are climatological mean, but the enhanced LW cooling of 32–36°N (Figure 9) is in the strong dust years. From Figures 8a and 8c (strong dust years), there are many dust aerosols in these regions.

*p. 12, ll. 14-15: The last sentence should be removed after the revision.*

Removed. Thanks for the catch.

*Trivial corrections*
*p. 1, l.14: "dust coupled" -> "dust-coupled"*
Done. (Page 1, Line 14). Thanks for the catch.

*p. 6, ll. 20-21: "were relatively low" -> "shows larger values"*

Done. (Page 6, Line 28). Thanks for the catch.

*p. 8, l. 15: "donwstream" -> "away", unless there is reason to define an atmospheric flow direction, which I may have missed*

Done. (Page 8, Line 20). Thanks for the catch.

*p. 10, l.7 "Dust direct radiative effects is" -> "Dust direct radiative effects for the atmosphere are"*

Done. (Page 10, Line 15). Thanks for the catch.

**References**

[revised manuscript text omitted]

---

## Author Response (AR3)

Manuscript No:acp-2017-55

Journal: ACP

The revised manuscript entitled "**Direct radiative effects of dust aerosols emitted from the Tibetan Plateau on the East Asian summer monsoon – a regional climate model simulation**" by Hui Sun, Xiaodong Liu, and Zaitao Pan.

*Editor (Comments to Author)*

*Thank you for your revised manuscript, which tried to address the issues raised by the Referees. The current version is greatly improved, but there are still a few rather minor issues.*

*More specifically, the most important one is that you should take care to avoid confusion between the terms "radiative forcing" and "radiative cooling rates", which are different and must be clearly differentiated to each other. You should use the appropriate terms in the paper. When discussing Fig. 9, you should refer to cooling rates (K/day), while discussing Fig. S2 you should refer to radiative forcings (W/m2). Please, look at the comments of the two Referees, address the raised issues and provide a further revised manuscript that should be accepted for publication in ACP.*

We thank the co-editor and the two reviewers for their valuable comments and helpful suggestions. We checked and corrected these terms throughout the manuscript to avoid confusion between the terms "radiative forcing" and "radiative cooling rates"; and we also addressed other issues identified by the reviewers in a point-by-point manner. In the following, the underlined italic texts are reviewer's comments and normal (font) texts are our responses. The bold texts have been inserted to new version of our manuscript.
* * *
*Referee# 1 (Comments to Author):*

*As a Referee #1 I have not access any more on the paper of Sun et al.. However, I have read through the reply to Referee #2 and I have some comments which I pass to you.*

*1) In their response they mention and added to the Discussion Section (Page10–11) the following:"One interesting finding of this study is the negative net (SW+LW) direct radiative forcing in the lower troposphere over the TP (Fig. 9)." Sorry but this is confusing. Fig. 9 does not show the net SW+LW radiative forcing but the net*

*SW+LW heating rate! It is often done throughout the text a confusion between heating rates and clear sky radiative forcing !*

Thanks for pointing this out. We checked the manuscript in its entirety thoroughly and corrected them in the latest version.

*In RegCM4, during the run and for a given time step, the radiation scheme is called two times, one with aerosol set to zero and one with aerosol (in this case dust aerosols). At each radiative time step, the instantaneous radiative forcing (at TOA and surface) are calculated respectively by the difference of TOA and surface net flux with and without aerosol. Then , if idirect=1 the flux and radiative temperature tendency (calculated as the divergence of the flux) from the No Aerosol call are passed back to the model, so the model dynamics and meteorology is not perturbed by aerosol in this case. If idirect=2, then the flux and radiative temperature tendency from the With Aerosol call are passed back and the dynamics is perturbed by aerosols.*

*The true instanteneous radiative forcing is obtained with idirect =1 because: a1) The With Aerosol and No Aerosol radiative calculation are made simultaneously with exactly the same atmospheric (T,h,p,q) profiles and surface albedo, and, a2) these profiles are not pertubed by aerosols during the run.*

*In this paper there are two 20-year simulations: b1) The first experiment is a control experiment (CON) with dust emitting sources both within and outside the TP. b2) The second experiment is a sensitivity experiment (SEN) where are turned off the dust emission in the northern and northeastern TP. I guess that the authors use idirect =2 for both experiments CON and SEN. Clear sky instantaneous Radiative Forcing is calculated for idirect=2 but take into account that this will be slightly different from the calculation with idirect=1 simply because you have some differentiation in atmospheric (T,h,p,q) profiles and dust load because of the dust aerosol feedback effect. Furthermore the calculated clear sky instantaneous radiative forcing is a different story from the clalculated SW and LW fluxes and heating rates simply because in these calculations clouds are present as well as the dust feedback effects on meteorological parameters. There are also semi-direct effects which are not discussed at all but could be also estimated.*

Thanks for providing the details information on the aerosol radiative treatment in RegCM4.1. We agree that it would be better to use idirect=1 for evaluating the true dust instantaneous radiative forcing, because the atmospheric (T,h,p,q) profiles and

surface albedo are not perturbed by aerosols during the run. However, if we had used idirect=1 in the experiments, we would have not obtain the dust radiative feedback to the climate since the main focus of this study is on the dust radiative feedback to the monsoon. Besides, dust emission from some sources may be overestimated by the lack of aerosol radiative feedback to atmospheric stability (Zhang et al., 2009). Therefore, it is more realistic to use idirect=2 in this study.

Yes, the model also includes semi-direct effects of dust aerosol, but interaction between dust aerosol and cloud is quite complicated, and the indirect effects of dust aerosol are not included in the model yet. Hence, the results on this may not be very robust and innovative in comparison to previous investigations. Especially, at the moment RegCM4.1 has not implemented indirect effects parameterization yet. We feel somewhat premature to assess the semi-direct effect that heavily involves clouds. We will explore these in our future work.

To address the reviewer's concern, we added the following sentence in the last paragraph of Discussion section (Page 12, Line 4–5). **The semi-direct effects (Hansen et al., 1997) are included in this study as part of atmospheric feedback, but are not explicitly discussed here since they would be better discussed along with the indirect effects as both involve clouds.**

*2) The authors mention "4. Although Fig. 9 shows a negative radiative forcing over the TP within the lowest 200-300 hPa of the atmosphere, the net atmospheric column forcing measured by the flux difference at surface and TOA, as normally done in the literature, is still positive (Fig. S2). Thus strictly speaking, by conventional definition, the direct net radiative forcing on the whole atmospheric column is still positive on long-term average (Fig. S2), even though the TP dust results in a cooling effect in the lower troposphere."*
*This is really confusing. Fig. S2 shows generally negative dust SW forcing and positive LW forcing at TOA and surface. My impression is that adding SW and LW forcing, still you get a negative radiative forcing for most of the domain both at SRF and TOA. I would suggest to add LW and SW forcings in Fig. S2 to see the total radiative forcing.*

The description is ambiguous. Our original meaning is that the SW absorbed by the whole atmospheric column (TOA minus Surface) is positive (increasing) on the long-term average over other source regions (similar to previous investigations),

despite the net atmospheric heating rate was negative over the TP in heavy dust years as a result of the radiative cooling effects of the dust aerosols. In the previous version, positive values actually refer to the difference (or sum depending on flux sign definition) between TOA and surface, not between SW and LW. At your suggestion, we plotted the sum of LW+SW as Fig. S2e (surface) and S2f (TOA). As you noted, they are mostly negative over the whole domain at surface and TOA, with surface cooling values being larger. We now clarified this as below (Page 11, Line 14–18).

**4. Although Fig. 9 shows a negative heating rate over the TP within the lowest 200−300 hPa of the atmosphere locally, the absorbed SW by atmospheric column measured by the flux difference between TOA and Surface, as normally done in the literature, is still positive outside TP over major source regions (Fig. S2b minus Fig. S2a). Thus strictly speaking, by conventional definition, the absorbed SW radiation is still mostly positive over the whole domain, even though the TP dust results in a cooling effect in the lower troposphere locally (Figs. S2e, f).**
* * *
*Referee# 2( Comments to Author ):*
*I am quite happy with the new submission by the authors. All my previous concerns have been addressed and I find the paper much improved and acceptable for publication. There are a few minor points that the authors may wish to take into account.*

We are very thankful to the reviewer for his/her thoroughness, rigorousness, persistence, and yet kindness in being considerate. We addressed these issues following the reviewer's suggestions. Please see below.

*Abstract*
*"The locally generated TP dust can cause surface cooling far downstream in eastern Mongolia ..." The authors may want to revise this statement after the new results show the effect on temperature being moved further east. Also the same phrase "...downstream in eastern Mongolia..." is mentioned in p. 9, l. 19.*

Thanks for pointing this out. We rephrased them as below (Page 1, Line 21, and Page 9, Line 14).

The locally generated TP dust can cause surface cooling far downstream in **Bohai Gulf and the China-North Korea border area** through stationary Rossby wave

propagation.

*p. 6, l 28: "...shows larger values..." I apologize for requesting this change in my previous review. It appears I missed that the authors refer specifically to the summer values. Please correct it back or rephrase it.*

We changed it back (Page 6, Line 28), thanks for the catch.

*p. 7, ll. 14-17: In my opinion, the authors identified the major cause of dominant LW cooling as the small dust particle size and the large ssa. The examination of day and night AOD differences had meaning in the effort to explain the LW cooling, but it now rather complicates the paper without showing significant differences. If the authors wish, they could remove this discussion here and simplify Fig. 6.*

We removed this discussion and simplified Fig.6 following the reviewer's suggestion.

*Fig. 9: Maybe the colorbar limits (minimum and maximum values) were not changed between this version and the previous one for comparison purposes in the review process. However for the final submission, the authors may consider redefining the limits.*

We redefined the limits in Fig. 9 following the reviewer's suggestion.

*Fig. 14: Although it is a nice addition to the paper, its format is not very helpful because the map is not visible. Please change it so that it conforms with the format of e.g. Fig. 3a..*

We replotted Fig. 14 following the reviewer's suggestion.

*p. 10, l 32: Papers with doi 10.3402/tellusb.v56i4.16439 and 10.5194/acp-12-7165-2012 also demonstrate the large sensitivity of SW radiative forcing to SSA.*

We read and cited them in the revised version (Page 10, Line 24–25).

*p. 13, ll. 2-3 : "The net atmospheric heating rate was negative over the TP in heavy dust years as a result of the radiative cooling effects of the dust aerosols..." The authors may want to clarify that they refer to the lower troposphere. Otherwise, there*

*is contradiction with their statement "Thus strictly speaking, by conventional definition, the direct net radiative forcing on the whole atmospheric column is ...positive on long-term average..."*

The description is indeed ambiguous. Our original meaning is that the SW radiation absorbed by the whole atmospheric column (TOA minus Surface) is positive (increasing) on the long-term average in other regions (similar to previous investigations), despite the net atmospheric heating rate was negative over the TP in heavy dust years as a result of the radiative cooling effects of the dust aerosols. We rephrased the paragraph as below (Page 11, Line 14–18).

**4. Although Fig. 9 shows a negative heating rate over the TP within the lowest 200−300 hPa of the atmosphere locally, the absorbed SW by atmospheric column measured by the flux difference between TOA and Surface , as normally done in the literature, is still positive outside TP over major source regions (Fig. S2b minus Fig. S2a). Thus strictly speaking, by conventional definition, the absorbed SW radiation is still mostly positive over the whole domain, even though the TP dust results in a cooling effect in the lower troposphere locally (Figs. S2e, f).**

[revised manuscript text omitted]

---

## Author Response (AR4)

Manuscript No:acp-2017-55

Journal: ACP

The revised manuscript entitled "**Direct radiative effects of dust aerosols emitted from the Tibetan Plateau on the East Asian summer monsoon – a regional climate model simulation**" by Hui Sun, Xiaodong Liu, and Zaitao Pan.

*Editor (Comments to Author)*
*thank you for your revised manuscript, which adequately addressed the comments of the Referees. It is now accepted for publication in ACP. The only change you should make is to remove the number (4) at the beginning of text in line 14, page 11 (page 17 of your uploaded Authors-response file), the number was placed by mistake.*

We removed the number (4) at the beginning of text in line 14, page 11 (the corresponding place of the uploaded Authors-response file) in the revised version. Many thanks.

[revised manuscript text omitted]
 locally, the absorbed SW by atmospheric column measured by the flux difference between TOA and Surface, as normally done in the literature, is still positive outside TP over major source regions (Fig. S2b minus Fig. S2a). Thus strictly speaking, by conventional definition, the absorbed SW radiation is still mostly positive over the whole domain, even though the TP dust results in a cooling effect in the lower troposphere locally (Figs. S2e, f).

Finally it is noteworthy that given the large variability of dust SSA among different sources in Asia, it is possible that the magnitudes or even signs of dust direct net radiative forcing on the atmosphere could vary among case studies and climate simulations over different continents. For example, the LW forcing of dust at Zhangye, China was found to be about a factor of two larger that over Saharan measured at Sal Island Cape Verde, owing to differences in the dust absorptive properties (Hansell et al., 2012).

It is very beneficial to study the impact of aerosols on climate using a RCM instead of a coarse-resolution global climate model (GCM). However, limited area RCM naturally cannot fully account for external forcing remote from the domain of interest although the lateral boundary conditions allow large-scale features to propagate into the domain. Our domain size (9600 km × 640 km) is reasonably large enough so that the weather and climate systems can have adequate spatial extent to develop within the domain, as attested by reasonable validation of wind pattern, temperature field, and precipitation (Section 3.1). Cautions should be exercised, however, that results from regional simulations could be somewhat domain-size dependent quantitatively although main results should not be affected. It is worth mentioning that the model's internal variability could influence the results; so we compared the standard deviation of summer surface temperature and precipitation in CON with the signal induced by the dust effects (CON minus SEN) during the heavy dust

years. The signal induced by the dust is much greater than the standard deviations (figures not shown). Therefore, the dust effect reported in our simulation is significant in the heavy dust years.

Only direct radiative effects of dust were included in our model and future studies should include both direct and indirect effects. The semi-direct effects (Hansen et al., 1997) are included in this study as part of atmospheric feedback, but are not explicitly discussed here since they would be better discussed along with the indirect effects as both involve clouds. 
[revised manuscript text omitted]

Hansen, J. E., Sato, M., and Ruedy, R.: Radiative forcing and climate response, J. Geophys. Res., 102, 6831–6864, 1997.

Hatzianastassiou, N., Katsoulis, B., and Vardavas, I.: Sensitivity analysis of aerosol direct radiative forcing in ultraviolet–visible wavelengths and consequences for the heat budget, Tellus B, 56, 368–381, doi: 10.3402/tellusb.v56i4.16439, 2004.

[revised manuscript text omitted]